# Unveiling the cut-and-repair cycle of designer nucleases in human stem and T cells via CLEAR-time dPCR

Nathan White [1], John Alexander Chalk[1,2], Yi-Ting Hu[1], Samuel Mark Pins[1], Chinnu Rose Joseph[3], Panagiotis Antoniou[3], Sandra Wimberger [3], Stina Svensson[1], Soraia Patricia Caetano-Silva[1], Anne Charlotte Adriane Mudde[1], Rajeev Rai[1], Sridhar Selvaraj[4,5], William Nelson Feist [4,5], Marianna Romito[1], Grzegorz Sienski [3], Roberto Nitsch [6], Claire Booth [1], Giorgia Santilli [1], Alessia Cavazza [1], Matthew Hebden Porteus [4,5], Marcello Maresca [3], Adrian James Thrasher [1] & Giandomenico Turchiano [1,2] ✉

DNA repair mechanisms in human primary cells, including error-free repair, and, recurrent nuclease cleavage events, remain largely uncharacterised. We elucidate gene-editing related repair processes using Cleavage and Lesion Evaluation via Absolute Real-time dPCR (CLEAR-time dPCR), an ensemble of multiplexed dPCR assays that quantifies genome integrity at targeted sites. Utilising CLEAR-time dPCR we track active DSBs, small indels, large deletions, and other aberrations in absolute terms in clinically relevant edited cells, including HSPCs, iPSCs, and T-cells. By quantifying up to 90% of loci with unresolved DSBs, CLEAR-time dPCR reveals biases inherent to conventional mutation screening assays. Furthermore, we accurately quantify DNA repair precision, revealing prevalent scarless repair after blunt and staggered end DSBs and recurrent nucleases cleavage. This work provides one of the most precise analyses of DNA repair and mutation dynamics, paving the way for mechanistic studies to advance gene therapy, designer editors, and small molecule discovery.

Recent advancements in designer DNA editors have demonstrated significant therapeutic potential for treating genetic disorders[1-5]. However, for the successful clinical translation of these tools in cell and gene therapies, a comprehensive understanding of their cellular activities is essential to enhance editing efficiency and ensure safety. Consequently, numerous techniques have emerged to detect or anticipate genotoxic events, reflecting an increasing demand for thorough characterisation of these genome-editing tools. Despite this progress, these detection methods present several limitations, including high costs, time-consuming protocols, the necessity for extensive bioinformatics expertise, and the risk of selective and sometimes biased evaluations of aberrations. Furthermore, the temporal dynamics of DNA editing and the subsequent repair processes remain poorly understood, leading to potential misinterpretations and observational biases.

Double-strand breaks (DSBs) are one of the most critical DNA lesions induced during genome editing and are typically resolved through non-homologous end-joining (NHEJ), alternative end-joining

[1]Infection, Immunity, and Inflammation Teaching and Research Department, Great Ormond Street Institute of Child Health, University College London, London, United Kingdom. [2]Cell and Gene therapy Safety, Clinical Pharmacology and Safety Sciences R&D, AstraZeneca, Cambridge, UK. [3]Genome Engineering, Discovery Sciences, BioPharmaceuticals R&D Unit, AstraZeneca, Gothenburg, Sweden. [4]Department of Pediatrics, Stanford University, Stanford, CA, USA. [5]Institute for Stem Cell Biology and Regenerative Medicine, Stanford University, Stanford, CA, USA. [6]Cell and Gene therapy Safety, Clinical Pharmacology and Safety Sciences R&D, AstraZeneca, Gothenburg, Sweden. ✉e-mail: g.turchiano@ucl.ac.uk

(alt-EJ), or homology-directed repair (HDR) pathways[6]. Programmable endonucleases, such as the clustered regularly interspaced short palindromic repeats-associated protein 9 (CRISPR-Cas9), leverage these repair mechanisms to achieve gene knockouts or, when combined with donor templates, facilitate the insertion of customised sequences into specific loci[7–9]. Given the potential for long-term therapeutic effects, targeted genome editing offers promising applications for a broad spectrum of genetic diseases[10]. However, unintended repair outcomes, such as indels, large deletions, and other substantial chromosomal aberrations, often hinder the desired editing effects[11–14] and pose significant genotoxic risks that could compromise both the safety and efficacy of these therapies[15].

To address these challenges, targeted integration enhancers (TIEs) have been developed by repurposing NHEJ and microhomology-mediated end-joining (MMEJ) inhibitors, such as AZD7648 and ART558, respectively, to promote template-mediated homology directed repair (HDR)[16,17]. While TIEs represent a promising strategy to minimise undesirable on-target DSB repair products, there is still a considerable lack of information regarding their impact on both on- and off-target aberrations.

Although many techniques have been developed to detect and quantify specific designer nuclease-induced aberrations, they fail to provide an absolute assessment of the frequency at which all undesired aberrations occur at on- and off- target sites. To quantify editing efficiency, most conventional methods rely on target site PCR amplification, such as sequencing based methods (e.g., ICE[18], CRISPRESSO[19]), enzymatic (e.g. T7E1[20]), or others (e.g. IDAA[21]; dPCR[22], rhAmpSeq[23,24]). While these strategies are effective for detecting small indel populations, they do not identify large deletions, DSBs, or other complex aberrations due to the inability of these sequences to be PCR amplified. Long-read sequencing offers greater detection capabilities for large deletions, but comes with higher costs, demands for specialised protocols and bioinformatic knowledge, and is still biased by kilobase spanning deletions[25–27]. These challenges are similar across various high-throughput, PCR-based techniques used to detect translocations (e.g., CAST-seq[28], HTGTS[29], UDITAS[30]) or genome-wide off-target effects (e.g., GUIDE[31], CHANGE[32], DIGENOME[33], BLISS[34], qDSB[35], DISCOVER-seq[36]). Alternatives that provide broader genomic characterisation, such as whole-genome sequencing, G-banding, FISH karyotyping, or comparative genomic hybridisation, sacrifice detection sensitivity for a wider overview[37]. Recent advances in imaging have led to the development of optical genome mapping[38], which bridges the resolution gap between karyotyping and NGS techniques on the kilobase scale (Supplementary Fig. 1a). By comparing the outcomes from various techniques, a clearer understanding of the mutational landscape can be achieved, although integrating these results remains challenging. Consequentially, the rate of endonuclease cleavage and subsequent DSB repair remain poorly characterised across cell types and gene targets due to the lack of a highly sensitive and quantitative assessment strategy[28,31,32,39,40]. Quantifying DSB induction over time has previously relied on partial or indirect measurements, such as repair product modelling from amplicon sequencing[41–43] or immunoprecipitation[44]. Furthermore, existing strategies have likely underestimated the frequency of precise DNA repair, and the time required for a cell to resolve a DSB. A thorough understanding of DNA cleavage and repair kinetics is essential for generating the safest and most effective gene editing platform for clinical purposes.

To measure fundamental editing processes, we develop a comprehensive assembly of multiplexed digital PCR (dPCR) assays termed CLEAR-time dPCR (*Cleavage and Lesion Evaluation* via *Absolute Real-Time dPCR*), a method previously referred to as MEGA dPCR (*Multipurpose Editing and Genotoxicity Assessment*), which reveals the prevalence of scarless DNA repair after DSBs, which subsequently leads to recurrent cleavage events by nucleases via dual normalisation after dPCR analysis.

CLEAR-time dPCR addresses critical gaps in the available genetic engineering analysis toolkit by providing a rapid, accessible, and specific overview of genome integrity post-gene editing that can be applied effortlessly to clinically relevant samples.

## Results

### CLEAR-time dPCR reliably quantifies nuclease-induced aberrations

By taking advantage of the ability of dPCR to quantify the absolute number of copies and linkage of DNA molecules[45,46], we developed a modular ensemble of multiplexed dPCR assays to quantify the integrity status of the DNA and its repair outcomes following delivery of CRISPR-Cas ribonucleoprotein (RNP) reagents to human primary CD34+ HSPCs (Fig. 1a).

**Wildtype, indels, and total-non-indel aberrations (Edge assay).** To quantify the fraction of loci harbouring wildtype sequences, indels, and total non-indel aberrations, an "Edge" assay comprised of a single pair of primers placed on either side of the RNP target site is used. This assay is complemented with a "cleavage" fluorescent-labelled oligo, i.e., a FAM probe placed directly over the prospective cleavage site, and a "distal" fluorescent-labelled oligo, i.e., a HEX probe placed ~25 bp up- or down-stream of the FAM probe (Fig. 1a; i). We observed a clear FAM/HEX double-positive population of droplets in mock electroporated cells, where no on-target cleavage (and therefore no mutations) was expected. In RNP-electroporated cells, the FAM cleavage probe does not efficiently bind to mutated amplicon sequences, resulting in an attenuated or total loss of FAM signal and, consequently, a drop in the number of amplified copies, thereby indicating indels. Non-indel aberrations such as large deletions, translocations, and unresolved DSBs at the on-target cleavage site completely disrupt amplification, resulting in a total loss of both the FAM and HEX copies relative to the reference assay (Fig. 1a; i, Fig. 1b; Supplementary Fig. 1b; i). The sum of Wildtype, indel, and other aberration populations encapsulates the entire edited cell population.

**Double-strand breaks, large deletions, and other structural mutations (Flanking and linkage assay).** The "Flanking" assay quantifies double-strand breaks, end processing, large deletions and other aberrations. It involves two amplicons flanking the cleavage site (5' and 3'), each with a probe nested within the primer pairs (Fig. 1a; ii). The linkage between these probed sequences is measured by the presence of double-positive signals within the same PCR droplet, normalised against the frequency expected by chance, as previously described[46]. In unedited cells, the Flanking assay predominantly yielded double-positive droplets, indicating minimal random DNA fragmentation from the genomic DNA isolation. Conversely, an increase in single-fluorescent positive droplets was observed in samples from RNP-electroporated cells, reflecting a decrease in linked sequences due to DSBs. Additionally, when the end processing involves both DNA strands on one or both sides of the cleavage, it prevents primer/probe binding, resulting in a loss of copies (Fig. 1b, Supplementary Fig. 1b; ii). It is important to note that the classification of large deletions is determined by the distance between the primer and probe relative to the cleavage site. Therefore, any DNA end processing greater than 20-30 bp from the cleavage site is classified as a large deletion, depending on the specific design of the assay.

**Aneuploidy assay.** Aneuploidy, i.e., full, or partial loss/gain of chromosomes, was quantified in this assay. Numerical variation of the p arm, q arm, or the whole chromosome results in the alteration of FAM, HEX, or both signals, whereas balanced segregated translocations could be observed as a gain of these signals. To quantify this, primers and probes are placed in the sub-telomeric regions of the p and q arms

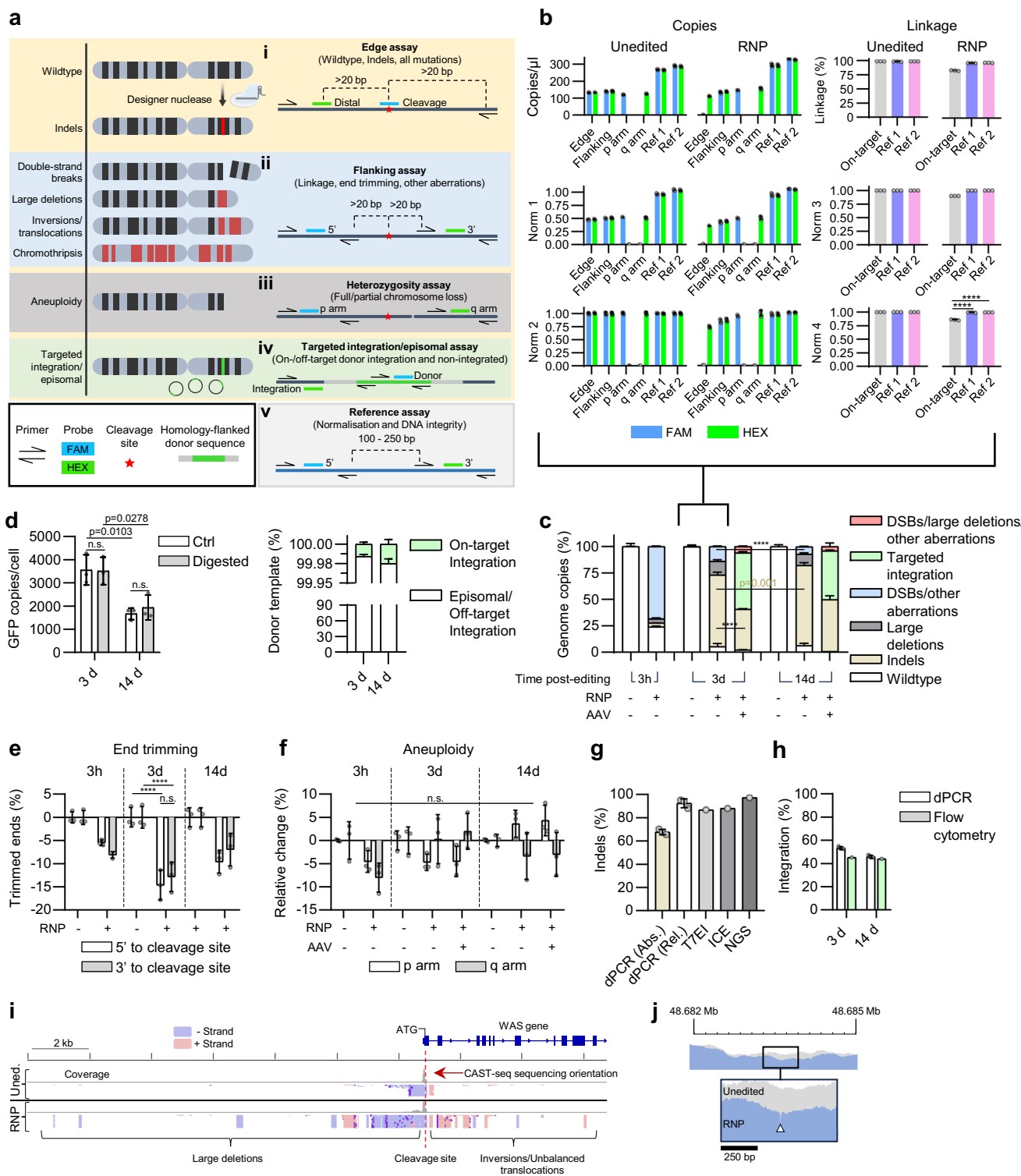

of the edited chromosome (Fig. 1a; iii, Fig.1b; Supplementary Fig. 1b; iii).

**Target-integrated and off target-integrated/episomal donor assessment.** The frequency of integrated and non-integrated templates when using a donor sequence for DSB correction via HDR was calculated using the "Targeted integration and episomal" assay. The assay consists of a primer bound to genomic sequence outside of the regions complementary to the donor homology arms, and a donor-specific primer, and a probe between them. A second set of primers is placed within the donor-specific sequence to detect all donor template copies, i.e., on- or off-target integrated and non-integrated episomes (Fig. 1a; iv, Fig. 1b; Supplementary Fig. 1b; iv, v). To evaluate donor DNA concatemers, an enzymatic digestion was performed within the donor sequence but outside the region covered by the second set of primers.

**Reference assays.** A pair of primers and probes on non-targeted chromosomes placed equidistant from those used in the flanking assays acts as a loading control and allows for copy and linkage normalisation to obtain an unbiased quantification of mutations identified by the mutation detecting assays (Fig. 1a; v, Fig. 1b; Supplementary Fig. 1b; vi).

**Fig. 1 | CLEAR-time digital PCR reliably quantifies nuclease-induced aberrations in HSPCs. a** Diagram illustrating the induced aberrations by designer nucleases and the CLEAR-time dPCR assay strategies. **b** Absolute copy number and linkage normalisation workflow. Data are shown as mean ± s.d. **c** Genome copy frequency summarisation of on-target aberrations 3 h, 3- and 14-days post-Cas9 editing with and without AAV transduction targeting the *WAS* locus. Data are shown as mean ± s.d. **d** Single cleavage restriction digestion of AAV donor template VCN and % integrated donor template measured in AAV-transduced cells 3- and 14-days post-editing. Data are shown as mean ± s.d. **e** End trimming was measured as absolute loss of 5′- and 3′- sequences flanking the Cas9 cleavage site 3 h, 3- and 14-days post-Cas9 editing in RNP-only edited cells. Data are shown as mean ± s.d. **f** Aneuploidy measured as the absolute change of p or q arm copy numbers. Data are shown as mean ± s.d. **g** Validation of indel frequency by comparing the relative indel frequency calculated by dPCR, T7EI assay, ICE and NGS measured in *WAS* edited HSPCs 3-days post-editing. Abs. and Rel. refer to absolute and relative indels (i.e., normalised or not normalised to a reference), respectively. Data for T7EI, ICE, and NGS represent *n* = 1, dPCR data shown as mean ± s.d. of *n* = 3 technical replicates. **h** Validation of donor-template integration by comparing digital PCR to flow cytometry of AAV-transduced cells at 3- and 14-days post-editing. Flow cytometry data represent *n* = 1, dPCR data shown as mean ± s.d. of *n* = 3 technical replicates. **i** Qualitative validation of large deletion and other aberrations of RNP only edited cells 3-days post-editing using CAST-seq. On top, *WAS* gene schematic, exons in bold. Light blue indicates aberrations on negative-strand, light red indicates aberrations on positive-strand (*n* = 2 technical replicates). **j** NGS targeted sequencing spanning ~2500 bp of the cleavage site targeting *WAS* (white arrowhead) indicating small and large deletions ( > 250 bp). X-axis indicates nucleotide position; Y-axis indicates number of mapped reads. Scale bar indicates 250 bp. All data represents *n* = 3 technical replicates unless stated otherwise. **b**, **c**, **e** One-way ANOVA with Sidak's multiple comparison test. **d** Two-way ANOVA with Tukey multiple comparisons test. **f** Two-way ANOVA with Sidak's multiple comparison test. n.s.= no statistical significance, ****p < 0.0001. Source data are provided as a Source Data file.

**Additional assays.** CLEAR-time dPCR can integrate various dPCR assays to enhance the detection of specific mutations around the target site. By designing custom sets of primers and probes based on knowledge gathered from sequencing-based detection techniques, it is possible to improve discrimination of particular mutations, such as microhomology-mediated end-joining (MMEJ) repair, translocations, or inversions. This not only enriches CLEAR-time dPCR's mutation discrimination capacity but also maintains the absolute character of the assessment, seamlessly integrating with the detection of other DSB repair products and reducing the quantity of mutations grouped as 'other aberrations'.

**Double normalisation.** A distinguishing feature of this methodology is the double normalisation of copy number and linkage, eliminating the biases derived from a relative or partial quantification (Supplementary Information 1). Briefly, each assay copy number is first normalised against averaged reference assays values. Subsequently, this initial ratio is normalised again, using values from the control unedited sample. This second normalisation accounts for inter-assay variabilities and accurately measures the effective variation, allowing quantification of absolute mutation population frequency (Supplementary Fig. 1c).

**Dynamic and unbiased chromosomal aberration analysis and validation.** To demonstrate the capabilities of CLEAR-time dPCR, primary human HSPCs were electroporated with a SpCas9/sgRNA RNP targeting the therapeutically relevant X-linked *WAS* gene, encoding the Wiskott-Aldrich syndrome protein. RNP electroporated HSPCs were split and either left as an RNP-only treatment or, by AAV transduction, delivered a donor-template encoding GFP flanked by 700 bp arms homologous to the RNP cleavage site[7]. Mock-electroporated cells were used as a control. Viable cells were counted (Supplementary Fig. 1d), and genomic DNA was then extracted 3 h, 3 days, and 14 days post-editing and analysed using CLEAR-time dPCR.

By applying the normalisations and summing the absolute locus frequencies, we created a unified plot for each condition. It is important to note that the retrieved data may not contain 100% of copies due to an accumulation of errors that exceed the logical limit. In this scenario, a careful review of setting dPCR droplet thresholds, an increased number of reference assays, or the exclusion of clear copy number outlier replicates may help reduce the error. Alternatively, an unexpected donor recombination and duplication or unusual structural rearrangements may be at the base of a biological reason.

Three h post-editing, the majority of other aberrations (68.4 ± 0.4%) were derived from DSBs with early evidence of indels and large deletions (Fig. 1c). After 3 days, DSBs mostly resolved as indels, though a small percentage of other aberrations persisted, likely representing a combination of unresolved DSBs and other gross chromosomal aberrations such as translocations or inversions. In AAV-transduced cells, donor-template integration is preferred mostly at the expense of indels, as observed by a significant decrease in the absolute percentage of indels between the RNP and RNP + AAV treatment groups (67.7 ± 1.9% and 38.4 ± 0.4%, respectively) (Fig. 1c). AAV-transduced cells also showed fewer large deletions and other aberrations compared to the RNP-only treatment group, albeit to a lesser extent than indels in absolute terms (Fig. 1c). Although large deletions remained stable at all time points post editing, the frequency of DSBs continued to drop to 7.0 ± 0.2%, indicating a continuous dynamic between DNA repair, loss of detrimental aberrations, and nuclease cleavage. The constant amount of wildtype loci would also reflect an ongoing nuclease activity 3 days post-editing and mainly later resolved as indels (Fig. 1c).

As the homology arms of the AAV donor-template are synonymous with the on-target *WAS* sequence, the flanking assay is unable to differentiate between large deletions and other aberrations when episomal AAV is present. However, with the TI assay we could instead track GFP copies in AAV-transduced cells 3 to 14 days post-editing (Fig. 1d). Although there was a 2-fold reduction of GFP copies between the two timepoints, the majority of the donor template molecules were still detectable ( > 99.4%), likely in the form of non-integrated episomes, and an even smaller amount likely integrated at off-target loci, as observed previously[47,48]. Additionally, digestion with a restriction enzyme with activity at a single cleavage site within potential concatemerized donor sequences did not result in a significant increase in VCN, suggesting these structures had not efficiently formed in the edited cells (Fig. 1d). DNA trimming of the 5′ and 3′ sequences flanking the cleavage site was dynamic, peaking at 3 days post editing at 14.6 ± 2.6% and 12.8 ± 2.6% for the 5′ and 3′ flanking sequences, respectively; however, there was no trimming bias in a particular direction (Fig. 1e). Conversely, there were no significant copy number deviations in chromosome sub-telomeric regions (Fig. 1f), indicating minimal aneuploidy.

To validate CLEAR-time dPCR, we aimed to benchmark and quantify double-strand breaks, indels, targeted integration, and chromosomal aberrations using orthogonal techniques. Digital PCR has been used to quantify double-strand breaks previously[43], however, this particular strategy results in the over estimation of DSBs due to the miscategorising of other aberrations, such as large deletions (Supplementary Fig. 1e).

The detection of indels demonstrated concordance across multiple techniques, including non-reference normalised dPCR, sanger and next-generation sequencing (NGS), and the T7EI assay (Fig. 1g; Supplementary Fig. 1f). However, since these methods are inherently biased, amplifying primarily intact sequences with few or no base insertions or deletions, CLEAR-time dPCR suggests that they overestimate indel frequency by approximately 17% in these samples. To

verify donor template integration, we performed flow cytometry on AAV-transduced cells and found that 45.3% of cells were GFP positive, which was comparable to the $53.4 \pm 1.01\%$ calculated by the targeted integration dPCR assay (Fig. 1h).

Lastly, to confirm the presence of large deletions and gross chromosomal aberrations, we employed CAST-Seq and 5 kb amplicon sequencing. Around the cleavage site, we identified inverted sequences and kilobase-spanning large deletions (Fig. 1i, j), providing qualitative diagnostic evidence for such events. The quantification of these events, as performed in previous publications[49], can suffer from significant biases due to larger deletions exceeding the distance of placed primers and gDNA extraction quality, thus skewing the interpretation. However, the qualitative approach used here complements the robust quantification achieved with CLEAR-time dPCR.

Taken together, these data show that the combined dPCR assays and appropriate normalisations provide a comprehensive, robust, and validated overview of the heterogeneous DSB repair products within the total post-editing cell population.

### CLEAR-time dPCR quantifies chromosomal aberrations and cell clonal dynamics in clinically relevant targets

We next sought to quantify the sensitivity of CLEAR-time dPCR and to demonstrate its broad-ranging applications. Clonal proliferation of cancerous cells from genotoxic by-products of gene therapy is a legitimate safety concern[50]. CLEAR-time dPCR can diagnose variations in the frequency of specific mutations within the edited loci at early stages, allowing for the prediction and tracking of cell clonal expansions. To determine the limit of detection (LoD) of a mutation within a population of wildtype sequences, we artificially recreated a large deletion event at the CCR5 target sequence, which occurs in HSPCs after Cas9 editing (Fig. 2a). We synthesised two reference linked cassette oligonucleotides: a "wildtype" and a 5' end truncated "large deletion" sequence mimicking that observed in the CCR5 gene (Fig. 2a; Supplementary Fig. 2a). By mixing the cassettes in decreasing "large deletion" to "wildtype" ratios and performing the dPCR flanking assay, we determined the LoD with this assay to be 2.33% (Fig. 2b).

To demonstrate the flexibility of CLEAR-time dPCR in evaluating the loci status of different gene targets, we electroporated HSPCs with RNP complexes targeting six therapeutically relevant genes with previously published gRNAs and tracked the consequences of DSB repair at these loci over time. To assess the contribution of HDR to loci repair, we also delivered a donor-template by AAV transduction following RNP electroporation for the BTK gene target (Fig. 2c; Supplementary Fig. 2b). Unlike the stable integration of the donor cassette via HDR in the WAS locus observed in AAV-transduced cells across timepoints, site-specific integration events at the BTK locus dropped by almost 2-fold from early to late timepoints and were replaced by indels (Fig. 2c; Supplementary Fig. 2c). The frequency of DSBs observed at 3 h post editing across the targets varied greatly from $18.5 \pm 0.5\%$ to $55.0 \pm 0.4\%$ (Fig. 2c). Indels marginally increased as other aberrations decreased between the 3-day and later timepoints (Supplementary Fig. 2d), whereas DNA end-trimming typically peaked at day 3 and different targets showed significant aneuploidy (Supplementary Fig. 2e).

It is noteworthy that across the various sites examined in this study, CCR5 (also considered a "safe harbour locus"[51]), EMX1, FANCF, RAG1, VEGFA, IL7R, CD34, and WAS (with the exception of BTK edited cells without a donor), we observe a stable or decreasing number of WT sequences while indels continue to increase beyond 72 h. Typically, DSBs delay cell proliferation through p53/p21 signalling[52]. Thus, cells exhibiting absent or reduced cleavage efficiency should theoretically possess a proliferative advantage, resulting in an increase in WT alleles over time, contrary to our observations.

These data strongly support the hypothesis of protracted nuclease activity, albeit significantly diminished, beyond 72 h post-electroporation.

An alternative hypothesis might consider the creation of mutations that confer a proliferative advantage. However, certain genes edited in this work, such as WAS, BTK, and FANCF, would rather hinder growth when mutated[53–55]. Furthermore, other sites examined, like VEGFA (which enhances vascularisation in tumours) and RAG1 and FANCF (which contribute to genomic instability), have not been previously associated with short-term proliferative advantages in vitro, unlike p53 or p21 knockouts.

Consequently, the most plausible explanation for these observations remains the persistent activity of the nuclease.

Notably, end-trimmed events, which consistently occur alongside indels, tended to disappear from the overall population over time, indicating that those mutations are associated with non-replication permissive aberrations or that the sequence is restored via homologous recombination with the sister chromatid as template.

### CLEAR-time dPCR detects aberrations across various designer nucleases and fine-tunes editor activity

To explore whether CLEAR-time dPCR is applicable across different cell types and designer nucleases, we electroporated PBMC-derived T cells with Cas9, Cas12a, or TALEN, each targeting the same position in the SH2D1A gene that encodes Slam-associated protein (SAP)[56]. Due to subtle variations in the precise cleavage site across the tested nucleases, we ensured that the 5' end of the FAM cleavage probe was located over the nucleotides most likely to form indels, as probe placement is integral for discriminating small indels (i.e., +1 insertions) from wildtype alleles (Supplementary Fig. 2f). We found a similar frequency of all types of aberrations induced by the different nucleases, again with no significant evidence of aneuploidy; however, cells edited with TALEN did show a 3' directional processing bias (Fig. 2d; Supplementary Fig. 2g). Amplicon sequencing revealed that only Cas9 produced +1 indels, which were clearly visible as a distinct droplet population in the dPCR dot-plot (Supplementary Fig. 2h).

Mitigation of the off-target cleavage activity of an RNP targeting the CCR5 gene when using a high-fidelity Cas9 has previously been described[28].To study the effect of Cas9 concentration on on/off-target cleavage activity, we edited cells at the CCR5 locus with decreasing concentrations of RNP and performed CLEAR-time dPCR at the cut site on chromosome 3 as well as at off-target sites on chromosomes 1, 13 and 19. At 3 h, we observed a clear pattern of decreasing DSB generation for both the on and off-target locus correlating with decreasing amounts of RNP, though notably we observe a significant decrease in off-target activity when reducing the RNP concentration to 1/8th of the original amount (Fig. 2e, f, Supplementary Fig. 2i). This pattern was observed again regarding indels three days post-editing; however, non-indel aberrations at the on-target site were present until the lowest RNP concentration (Fig. 2e, f). These results demonstrated the effective utilisation of CLEAR-time dPCR to determine the optimal nuclease concentration for targeted cells, thereby mitigating unwanted effects and maintaining high on-target efficiency.

### HSPC aberration landscape follow-up in longitudinal murine studies

Recent pre-clinical studies have consistently observed a significant decrease in genome-edited HSPCs harbouring the intended therapeutic sequence post-transplantation[57]. We corroborated this phenomenon by screening gene edited transplanted cells with CLEAR-time dPCR and further characterised the mutation dynamics occurring after xenotransplantation. We obtained human HSPCs in which the gene encoding X-linked inhibitor of apoptosis protein (XIAP) had been targeted with SpCas9, transduced with an AAV encoding either a GFP- or codon-optimised XIAP sequence or GFP with homology arms

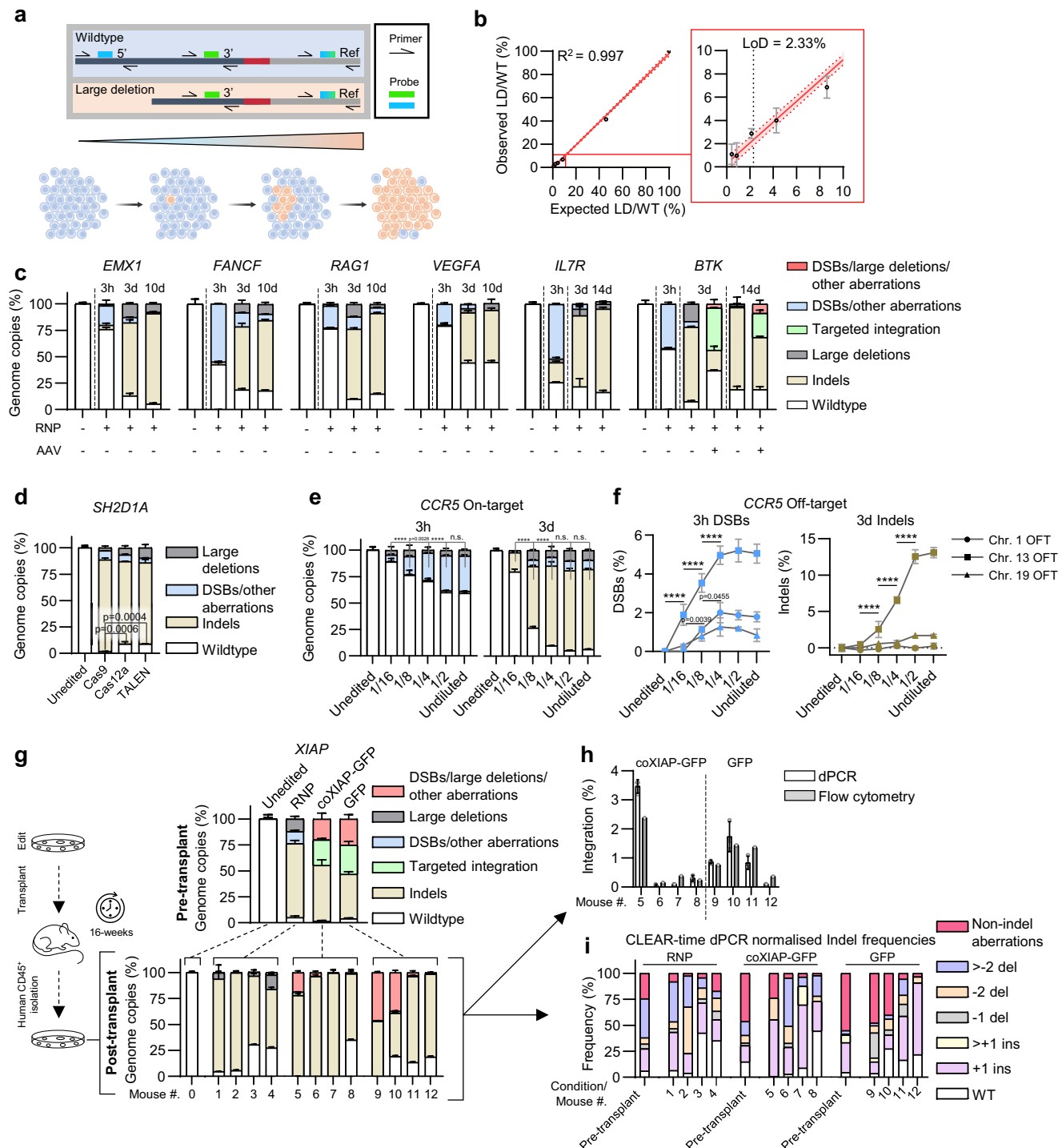

synonymous with the target site, and transplanted into immunodeficient NOD.Cg-Prkdcscid Il2rgtm1Wjl/SzJ NSG mice (Fig. 2g).

CLEAR-time dPCR revealed that prior to transplantation in AAV-transduced cells, indels were again found to be reduced at the expense of integration, with an integration frequency in AAV-transduced cells found to be ~25% (Fig. 2g; Pre-transplant). Sixteen weeks post-transplantation, the aberration frequencies showed a large degree of variability, with evidence of skewed dominance of mutant sequences (Fig. 2g; Post-transplant). As expected, alleles carrying the template sequence were significantly reduced post-transplantation. This was confirmed by flow cytometry (Fig. 2h; Supplementary Fig. 2j). To resolve whether there was clonal proliferation of a particular indel repair product, we sequenced the post-transplant samples and normalised the absolute population frequencies using CLEAR-time dPCR. We found an overrepresentation of +1 and −2 indels compared to the cells initially transplanted into the mice (Fig. 2i). Additionally, cells derived from two mice that received HSPCs edited with the GFP control vector showed a large increase in non-indel aberrations as opposed to indels (Fig. 2i). Together, these data illustrate that CLEAR-time dPCR has valuable roles in all stages of gene therapy development, from early therapy development to pre-clinical models and post-clinical follow-up once administered to patients and combined with orthogonal methods will further the characterisation in the repair outcomes.

**Fig. 2 | CLEAR-time dPCR detects chromosomal aberrations and cell clonal dynamics in clinically relevant targets in vitro and in vivo. a** Schematic of wildtype and large deletion cassettes used to establish the LoD of large deletions, and representation of clonal expansion of cells harbouring a large deletion (orange) mediated genotoxic aberration amongst wildtype cells (blue). Dark and light grey bar indicates *CCR5* and reference sequences, respectively. Red bar indicates flag sequence used to fuse assay and reference sequences. **b** Correlation of observed against the expected percentage of large deletions. Vertical black dotted line represents the limit of detection. Solid and dotted red line represents line of regression with 95% confidence interval, respectively. $R^2 = 0.997$. Calculated with linear regression analysis in GraphPad. Data shown as mean ± s.d. of $n = 4$ technical replicates. **c** CLEAR-time dPCR summaries of Cas9-edited HSPCs targeting various genes at different timepoints. The *BTK* edited HSPCs were also transduced with AAV6 encoding GFP. All editing was normalised against unedited mock electroporated HSPCs. Data are shown as mean ± s.d. **d** CLEAR-time dPCR summary of *SH2D1A* edited T cells with Cas9, Cas12, and TALENs at 3 days post-editing. Data are shown as mean ± s.d. **e** CLEAR-time dPCR summary of on-target *CCR5* edited with decreasing concentrations of Cas9 at 3 h and 3 days post-editing. Data are shown as mean ± s.d. **f** DSB and indels quantification at three known off-targets targeting *CCR5* with decreasing concentrations of Cas9 at 3 h and 3 days post-editing. Data are shown as mean ± s.d. **g** CLEAR-time dPCR on *XIAP* edited HSPCs pre-transplant and 16 weeks post-transplant. Data are shown as mean ± s.d. $n = 4$ mice for each treatment group. **h** Integration frequency in 16-week post-transplant *XIAP* edited and AAV-transduced hCD45 cells by dPCR and flow cytometry. Flow cytometry data represent $n = 1$ per mouse, dPCR data shown as mean ± s.d. of $n = 3$ technical replicates. **i** CLEAR-time dPCR normalised ICE analysis of pre-transplant *XIAP* edited HSPCs and post-transplant *XIAP* edited hCD45 cells. All data represents $n = 3$ technical replicates unless stated otherwise. **d**–**f** Two-way ANOVA with Tukey post-hoc test. n.s.= no statistical significance, ****$p < 0.0001$. Source data are provided as a Source Data file.

## Targeted integration enhancers decrease genome stability in HSPCs, iPSCs, and T cells

Integrating a template sequence at a specific location using HDR for gene therapy has significant advantages, but its effectiveness is hindered by indel formation through the NHEJ or Alt-EJ repair pathways, which are semi-dependent on the cell cycle[58–60]. To mitigate this, small-molecule mediated inhibition of end-joining pathways has been implicated in increasing template-mediated HDR-integration, in some cases by greater than 50-fold[16,17]. Nevertheless, the characterisation of systemic inhibition of end-joining repair pathways and off-target activity in a gene-editing context has not yet been fully explored. To address this, we cultured CCR5 Cas9-edited HSPCs in the presence of DSB repair-inhibiting compounds targeting NHEJ and MMEJ pathways. For NHEJ, we used AZD7648, a potent inhibitor of the DNA-PK protein, which plays a pivotal role in the repair cascade by facilitating recruitment, coordination, and activation of necessary components. For MMEJ, we employed ART558 or PolQi2, which inhibit POLQ, a key enzyme involved in DNA end processing, helicase activity, and ends alignment. In some experiments, we used a combination of both types of inhibitors. Prior to the removal of the targeted integration enhancers (TIEs) at 24 h, we observed a 2-fold increase in DSBs and up to an 8-fold decrease in indels relative to the RNP-only treatment (Fig. 3a). DSBs in all treatment groups were mostly resolved by two weeks, with an unexpectedly large increase in wildtype loci in the AZD7648 treatment groups (Fig. 3a).

The relevance of adopting a more comprehensive and relatively quick evaluation such as CLEAR-time dPCR is exacerbated at the day 1 post-editing data point in Fig. 3a. Results relying solely on sanger sequencing (or other common methods such as deep sequencing, enzymatic digestion or relative dPCR strategies) may lead to the erroneous hypothesis that the drugs interfere with Cas9 activity. This misinterpretation arises when observing over 95% of WT loci with DNA repair inhibitors, compared to less than 25% WT sequences in HSPCs cultured in typical circumstances. This discrepancy occurs despite uniform electroporation of cells, which were subsequently divided into different media conditions.

In contrast, CLEAR-time dPCR resolves this matter, revealing less than 5% (instead of >95% by sanger sequencing) WT sequences in the same samples treated with drugs (Supplementary Fig. 3a). This example highlights an alternative, more accurate approach to analysing the activity and effects of designer editors on targeted regions. By eliminating these biases, CLEAR-time dPCR prevents the drastic miscategorisation of over 95% of the assayed loci.

Next, we found that up to 50% of sequences were trimmed in both the 5′ and 3′ directions relative to the cleavage site prior to the removal of the compounds and persisted at a low frequency after 2 weeks (Fig. 3b).

We next explored TIEs in a therapeutic context by knocking in a single-stranded DNA donor template (ssODN) into the *CD34* locus in clinically relevant cells. In induced pluripotent stem cells (iPSCs), we observed a significant increase in large deletions, DSBs/other aberrations, and end trimming when cultured with TIEs; however, this was rescued when co-electroporated with an ssODN carrying homology arms of about 35 bases to the relative target site (Fig. 3c; Supplementary Fig. 3b). Conversely, with HSPCs, we observed up to 20% of end-trimmed sequences with TIEs that were not rescued when co-electroporated with an ssODN (Fig. 3d; Supplementary Fig. 3c, d). This difference between the cell types tested is likely due to the relatively higher HDR efficiency (up to 89.2 ± 0.7%) achieved in iPSCs compared to HSPCs. Next-generation sequencing of these samples shared the same relative quantification bias as ICE, with up to 50.1 ± 1.9% of loci unaccounted for (Supplementary Fig. 3e, f). To investigate how CLEAR-time dPCR compares to previously published literature using TIEs[17], we obtained samples of Cas9-edited and AAV-transduced T cells, HSPCs, and iPSCs that were cultured with AZD7648. We found that up to 30.2 ± 1.8% of loci were unaccounted for due to DSBs, other aberrations, and large deletions (Fig. 3e; Supplementary Fig. 3g). Together, these results further substantiate the issues related to bias inherent in using relative quantification methods, which CLEAR-time dPCR effectively eliminates. Additionally, our findings corroborate previous reports that inhibiting the repair of DSBs enhances targeted integration.

## Kinetics analysis reveals NHEJ precise repair, mutation likelihood, and recurrent nuclease cleavage

Editing activity is influenced by numerous factors, including, but not limited to; sequence of the target locus, cell type, designer editor and its method of delivery, presence/type of a donor repair template, and culture conditions[56,61–63] (Fig. 4a). As CLEAR-time dPCR detects and distinguishes DNA status and repair without evident biases, we endeavoured to clarify fundamental biological characteristics of on-target designer nuclease activity and the cellular response to DSBs. To elucidate dynamics such as the rate of aberration formation, DSB half-life, and repair product likelihood, we conducted multiple time series experiments with different cell types, nucleases, and gene targets. To gain novel insights into cellular repair activities like the frequency of precise (error-free) DSB repair and recurrent nuclease cleavage, we edited cells in each experiment with RNP and cultured half in DSB repair inhibitors (TIEs: AZD7648 and ART558).

Initially, we electroporated HSPCs with a newly described high-fidelity Cas9 variant, PsCas9[64], complexed with an sgRNA targeting the gene encoding *CD34*. Genomic DNA was collected at multiple time-points up to 14 days post-nucleofection and analysed using the Edge, Flanking, and Reference assays. TIEs were removed from the cells after 24 h to replicate the therapeutic strategy of most ex-vivo approaches, which freeze cells 1 or 2 days post-treatment[16], and to also maintain cell viability for the remaining timeseries extractions. Based on the dPCR timeseries, the greatest DSB frequencies occurred between 4–12 h and

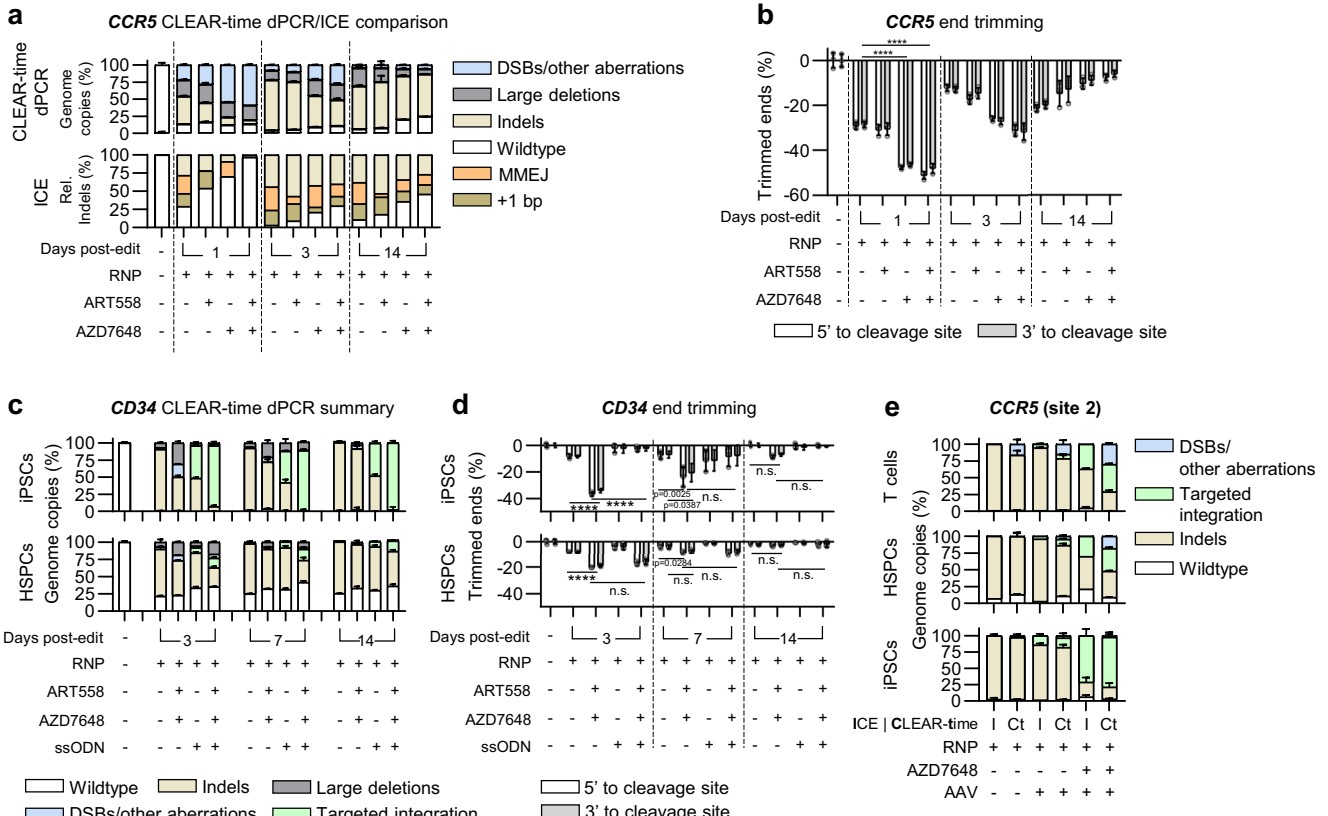

**Fig. 3 | Targeted integration enhancers decrease genome stability in HSPCs, iPSCs, and T cells.** Characterisation of aberrations induced by repair inhibitors on Cas9 edited HSPCs targeting the *CCR5* locus. **a** Summary comparison between CLEAR-time dPCR (top) and ICE (bottom) 1, 3, and 14-days post-editing. Rel. indels = relative indels. For CLEAR-time dPCR, data are shown as mean ± s.d. For ICE, data are shown as $n = 1$. **b** Quantification of sequence trimming around the cleavage site in *CCR5* edited HSPCs 1, 3, and 14-days post-editing. Data are shown as mean ± s.d. **c** CLEAR-time dPCR summary of aberrations induced by repair inhibitors on Cas9 edited iPSCs (top) and HSPCs (bottom) targeting the *CD34* locus with and without an ssODN 3, 7, and 14-days post-editing. Data are shown as mean ± s.d.

**d** Quantification of sequence trimming around the cleavage site of *CD34* edited iPSCs (top) and HSPCs (bottom) 1, 3, and 14-days post-editing. Data are shown as mean ± s.d. **e** Comparison between ICE (I) and CLEAR-time dPCR (Ct) in Cas9 edited T cells (top), HSPCs (middle), and iPSCs (bottom) targeting the *CCR5* locus with and without AAV transduction and AZD7648 treatment 4-days post-editing. Data are shown as mean ± s.d. ($n = 3$ technical replicates of 2 independent donors per cell type). All data represents $n = 3$ technical replicates unless stated otherwise. **b**, **d** Two-way ANOVA with Tukey post-hoc test., n.s.= non-significant, ****$p < 0.001$. Source data are provided as a Source Data file.

began to resolve as indels and large deletions within the first 24 h. The wildtype population declined faster in the first 8 h when cultured with TIEs, but both treatments reached a consistent frequency of 14.9 +/− 1.0% by 12 h that remained stable through the end of the experiment. TIEs also delayed the formation of all aberrations, leading to an accumulation of more DSBs, which quickly resolved after their removal at 24 h (Fig. 4b). Repair with indels, on the other hand, continued for 4 days, with a slightly lower frequency observed in repair-inhibited cells. In accordance with previous test, large deletions were observed to be higher with TIEs and to decrease over time giving space to small indels but not to wildtype loci (Fig. 4b).

To infer continuous frequencies and derive rate coefficients, we modelled nuclease cleavage and subsequent DSB repair utilising an updated version of the previously described three-state model of Cas9 kinetics[41,42] (Fig. 4c). In this model, the RNP traffics into the cell post-electroporation over an initial delay window (t) to locate and cleave its target sequence based on a cleavage rate coefficient $k_{dsb}$. The DSB is identified and resolved by cellular repair machinery into an indel, large deletion, or TI product based on corresponding rate coefficients $k_{in}$, $k_{ld}$, or $k_{ti}$, respectively. Through NHEJ or HDR, the cell can also precisely repair the cut site ($k_{pr}$), restoring the wildtype sequence which can then be re-identified and cut by the RNP. As removal of TIEs fundamentally alter cellular repair dynamics, ordinary differential equations (ODEs) were fit to the kinetics data obtained from the first 24-h of each

timeseries prior to their removal. All fitted genome copies frequency curves were validated by correlating the model against the observed data and were found to have a high goodness of fit with coefficient of determination ($R^2$) ranging between 0.95 and 0.99 across loci types (source data).

From the fitted curves, we were able to extrapolate the frequency at which aberrations occurred at any given time (Fig. 4d, Supplementary Fig. 4d). Modelling revealed that DSBs peaked with a frequency of 38% at 3.9 h post-electroporation. This peak was delayed with TIEs to 6.5 h, but at a frequency of 73.4%. Using the derivatives measured from the ODEs, we calculated the maximal velocity ($V_{max}$) of DSB generation in repair inhibited cells to occur at 75 minutes post-electroporation at a rate of 0.43% loci min[-1] (Fig. 4e). The time for indels and large deletions to reach $V_{max}$ occurred concomitantly with the peak of DSB event frequency regardless of repair inhibition, reflecting their dependence on available DSB substrate.

As precisely repaired loci cannot be directly distinguished from unedited wildtype sequences, we derived precise DSB repair by comparing the wildtype and DSB events through the repair-inhibited samples. We hypothesized that the rapid reduction in wildtype frequency with TIEs in the first 12 h was not due to the repair inhibitors increasing nuclease cleavage efficiency, but rather, they were preventing the cellular precise repair mechanics from occluding the true extent of cutting activity and recurrent target cleavage. It follows that

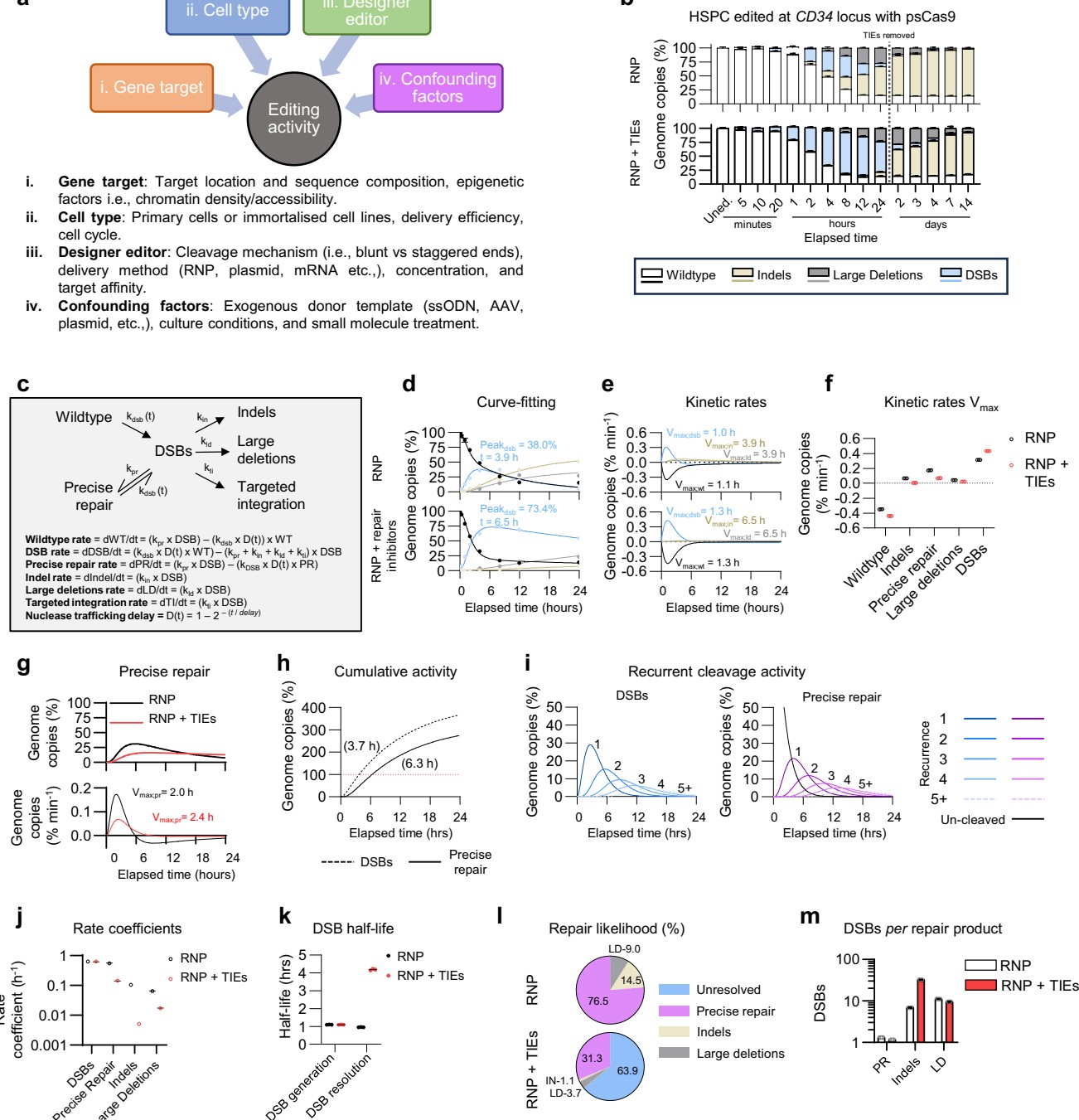

**Fig. 4 | Precise repair and recurrent cleavage activity of PsCas9 in HSPCs.**
**a** Overview of factors affecting nuclease-based editing activity. **b** Stacked bar-charts of observed dPCR timeseries genome copy frequency summaries of *CD34* edited HSPCs using PsCas9 with and without repair inhibitors 5 minutes to 14 days post editing. Data are shown as mean ± s.d. **c** Updated three-state model of nuclease-induced cleavage and subsequent precise or mutation repair. Where; $k_{dsb}$, $k_{pr}$, $k_{in}$, $k_{ld}$, $k_{ti}$ = rate coefficients of DSBs, precise repair, indels, large deletions, and targeted integration, per h, respectively. D(t) = Cas9 nuclear trafficking delay based on the time in hours. **d** ODE fitted curves (lines) modelled using the first 24 h of CLEAR-time dPCR data. Dots and error bars indicate mean ± s.d. of observed dPCR data. **e** Kinetic activity rates per minute derived from the ODE fitted curves. $V_{max}$ = time at which maximum event velocity occurred. **f** Comparison of event $V_{max}$ in untreated and DSB repair-inhibited cells. **g** Comparison between untreated and repair-inhibited HSPCs precisely repaired genome copy frequency (top) and rate of precise repair (bottom). **h** Accumulation of DSBs and precisely repaired DNA in RNP-only edited cells in a 24-h time-period. Values in brackets indicates time at which

DSBs exceeded 100% genome copies. **i** Model of recurring cleavage activity of precisely repaired DSBs. Blue and purple lines indicate frequency of DSBs and precise repairs, respectively. Black line indicates uncleaved sequences.
**j** Comparison of rate coefficients, in events/h, estimated by the ODE kinetics between untreated and repair-inhibited cells. **k** Comparison of average half-life of DSBs to generate from uncleaved sequences and resolve as a repair product in untreated and repair inhibited HSPCs. **l** Pie-charts of the likelihood of DSB resolution into each measured repair product or remaining unresolved per hour. Calculated using the ratio of rate coefficients and the sum of all repair outcome rate coefficients. **m** Average number of DSBs necessary to generate each repair product. Calculated using the ratio of the specific repair product rate coefficient and the sum of the rate coefficient of all repair products. All data represents *n* = 3 technical replicates unless stated otherwise. All data derived from modelling (*n* = 3 replicates, 1000 bootstraps/kinetic) is shown as mean ± s.d. Source data are provided as a Source Data file.

the observed difference in the Wildtype population between treatment groups reflects the difference in the cell's capacity to precisely repair DSBs. Thus, the model derived a single $k_{dsb}$ rate coefficient from both RNP-only and repair-inhibited conditions and generated two $k_{pr}$ coefficients to capture the observed rate differences. We verified the accuracy of our $k_{pr}$ values by comparing the deltas of our modelled precise repair loci over time against the deltas from the fitted curves of the observable wildtype alleles (Supplementary Fig. 4a).

We found that the $V_{max}$ of precisely repaired loci occurred earlier than loci repaired with a mutation (2.0 h vs ca. 3.9 hrs) (Fig. 4f, g), illustrating the relative speed of error-free repair of damaged DNA. The frequency of precisely repaired loci peaked at 31.3% at 5.8 h (Fig. 4g) but noticeably declined from this point indicating that the nuclease was still active and continuing to cleave the regenerated "wildtype" sequences. To explore this phenomenon of recurrent cleavage and repair activity, we used the ODEs to calculate the total accumulated number of DSB events and precisely repair loci generated over the 24-h timeseries. Indeed, both DSB events and precisely repaired loci exceeded 100% of genome copies within 3.5-6.5 h and reached values of ~360% and ~270% by 24 h respectively, suggesting that on average, wildtype loci experienced ~3.6 rounds of cutting and were precisely repaired 2.7 times (Fig. 4h). We modelled this recurrent activity by recursively applying the rate coefficients to the progressively reducing pool of cleavable sequences as schematized in Supplementary Fig. 4b.

In this model, DSBs events and precisely repaired loci were each grouped based on the number of recurrent cleavages. We found that by 24 h, a minimum of 5 cycles of cleavage events were likely to have occurred in progressively smaller subsets of loci, and that by 12 h the number of uncleaved sequences depleted below 1% (Fig. 4i). The calculated rate coefficients, i.e., the relationship between loci concentration and its formation rate per h, allowed us to derive additional information regarding nuclease and cellular repair activities. We found that, even with DSB repair inhibition, loci were still repaired in a precise manner, though, like indels and large deletions, the rate was significantly slower (Fig. 4j). The indel generation coefficient appeared to be particularly affected by the TIEs, dropping from 0.105 h$^{-1}$ to 0.005 h$^{-1}$. Moreover, we found the half-life of DSB generation and resolution, i.e., the time required for half of the total sequences to be cleaved or repaired, to be 1.1 and 1.0 h, respectively (Fig. 4k). This demonstrates further the potential for repair activity to occur just a few minutes after DSB generation, as supported by the observed and modelled kinetics data (Supplementary Fig. 4c). Notably, in the presence of repair inhibitors, the average half-life of DSB resolution increases up to 4.2 h. Using the rate coefficients, we next calculated the likelihood of each repair product forming, or left unresolved per h, for a DSB at any given timepoint in the 24-h post-editing window. Resolution through precise repair was the predominant product being formed after cleavage regardless of DSB repair inhibition, whereas, unsurprisingly, the majority of DSBs were unresolved when DSB repair was inhibited (Fig. 4l). In addition to the repair likelihood, we provide an estimate of the average number of double-strand breaks likely to have occurred before a particular repair product is formed. In these experimental settings, we determined that it takes on average ~1.3, 6.9 or 11.1 unique DSB events to resolve as a precisely repaired, indel, or large deletion, respectively (Fig. 4m). The number of DSBs to occur, before indel repair products were generated, increased by approximately 5-fold when repair was inhibited, with indels taking ~30 DSB events to generate (Fig. 4m). Together, these data show that Cas9-induced DSB and repair kinetics can be quantified using CLEAR-time dPCR with unprecedented accuracy in primary cells and for clinically relevant protocols, and modelling these dynamics with the inclusion of repair inhibitors elucidates specific designer nuclease activity and cellular repair mechanisms. Continued investigation employing temporally controlled nucleases, such as vfCRISPR[41] or CRISPRoff[65], will further elucidate the complexities of DNA repair kinetics.

## Comparative DNA Repair Dynamics: SpCas9 blunt vs. PsCas9 staggered End cleavage

To elucidate further biological information, we next compared the cleavage activity between SpCas9, a blunt-end cutter, and PsCas9, a staggered-end cutter, in K562 cells targeting *CD34* with an ssODN donor template. Similar to the HSPCs, we observed more DSBs and a delay in aberrations forming when cells were cultured with repair inhibitors.

Interestingly, the targeted integration across the different treatment starts to significantly increase only after the DNA repair inhibitor removal from the media indicating that it is possible to better time the presence of Donor templates and DNA repair inhibitors preserving higher targeted integration with cell viability and genomic stability (Fig. 5a).

The overall cleavage activity of PsCas9 was less efficient than SpCas9 as evidenced by a slower declining of WT loci and ascending DSBs in cells edited with SpCas9 (Fig. 5b and Supplementary Fig. 5a–e). The discrepancy of the cleavage efficiency is reflected by the time taken for half of the wildtype sequences to be cleaved increasing from 2.5 h to 8.3 h in SpCas9 and PsCas9 edited cells, respectively, while the DSB resolution half-life was showed to be at 2.3 and 2 h on average (Fig. 5c). Again, we found the most common type of repair product for both nucleases being a precisely repaired locus though this was more prevalent in PsCas9 edited cells when cultured with DSB repair inhibitors (Fig. 5d). This was also reflected in the number of DSBs to have occurred prior to repair product formation, indicating subtle differences depending on the DNA cleavage efficiency (Fig. 5e). Despite the variation in cleavage patterns, blunt versus staggered, the DNA repair rate coefficients, which are independent of the DSB generation rate, remain comparable for both nucleases in this side-by-side test (see Supplementary Fig. 5a–e). This indicates that the DNA repair mechanisms are not influenced by these two different types of cleavage products.

## Comparative DNA repair dynamics: primary T cells vs. K562 cell line

We next explored the editing kinetics of SpCas9 targeting the same locus (*WAS*) in primary (PBMC derived T cells) and an immortalised cell line (K562) transduced with a GFP-encoded AAV. Most HDR integration activity occurred within three days post-editing, with TIEs nearly doubling the HDR efficiency in both primary cells and cell lines (Fig. 5f). When modelling these datasets, the rate of targeted integration was, in fact, the slowest repair product, showing a seven-fold decrease in $V_{max}$ relative to indel generation in the first 24 h (Fig. 5g, Supplementary Fig. 5f–j), reflecting much slower kinetics for homologous recombination (HR) relative to other repair pathways. Interestingly, the cleavage efficiency of SpCas9 was comparable between the cell types, indicating minimal epigenetic influences on nuclease accessibility at this particular locus (Fig. 5h).

In both cell types we again observed that a precisely repaired locus was the most prevalent repair product, even when cultured with TIEs in the presence of a donor template (Fig. 5i) and, as expected, K562 returned higher rate of targeted integration in respect to the T-cells. It is noteworthy that in the side-by-side test and these cell type comparisons, the number of DSBs required to achieve targeted integration decreased when NHEJ repair inhibitors were used. This finding suggests that even if the HDR pathway does not accelerate (same rate coefficient values with and without the inhibitors), the reduction of NHEJ leaves DSB substrate available for an extended period favouring repair by targeted integration (Fig. 5e, j).

Together, these data show that DNA cleavage by designer nucleases and subsequent repair are highly dynamic and are influenced by cell type, locus, and presence of a donor template.

Finally, to compare the values obtained from different nucleases and cell types under different experimental conditions and to

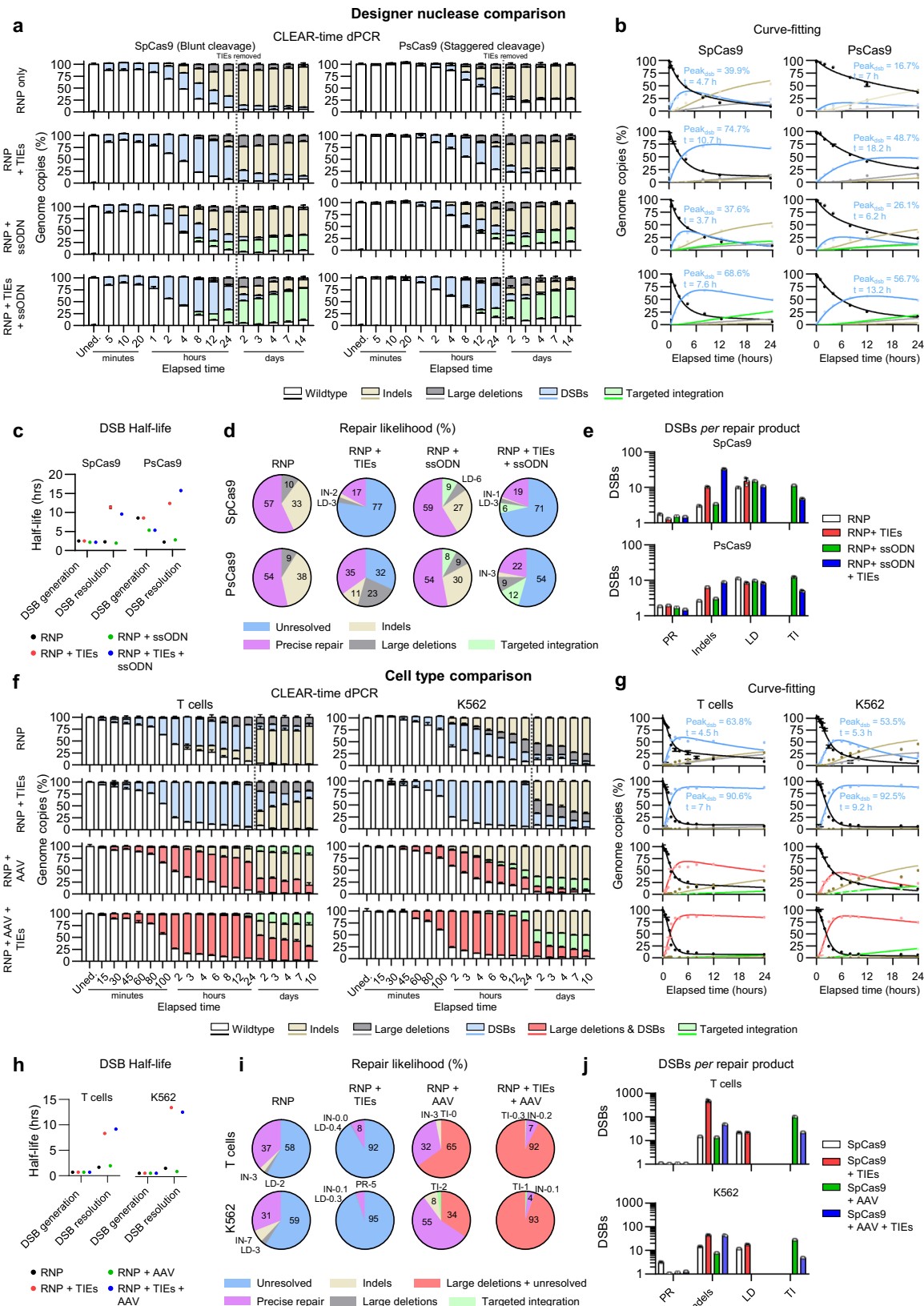

summarise our findings, we utilised rate coefficient parameters from each kinetics experiment. These parameters, independent of cleavage efficiencies, illustrate the rates at which each mechanism involved in DSB induction and subsequent repair occurs. Previous attempts to resolve these numbers employed various methods[41,42,66]; here, we report and compare the outcomes, highlighting differences in quantity

and quality of evaluations by also considering large deletions, HDR, and an unbiased assessment of precise repair that aligns with the expectations as the main repair outcome following DNA damage (Fig. 6).

By using CLEAR-time dPCR combined with DSB repair inhibitors to delineate cleavage and repair activity, our approach not only

**Fig. 5 | Designer nuclease cleavage and repair kinetics in human primary cells and cell lines. a** A stacked bar chart of observed dPCR timeseries genome copy frequency summaries of edited K562 cells using SpCas9 (blunt cut) and PsCas9 (staggered cut) with and without repair inhibitors/ssODN donor sequence, 5 min to 14 days post editing. Media change 24 h represented with vertical dotted bars. Data are shown as mean ± s.d. **b** ODE fitted curves (lines) modelled using the first 24 h of CLEAR-time dPCR data on SpCas9 (left) and PsCas9 (right) edited cells. Dots and error bars indicate mean ± s.d. of observed dPCR data. **c** Comparison of the average half-life of DSBs generated from uncleaved sequences and resolved as a repair product in untreated and repair-inhibited HSPCs. **d** Pie-charts of the likelihood of DSB resolution into each measured repair product or remaining unresolved per hour. **e** Average number of DSBs necessary to generate each repair product. Data are shown as mean ± s.d. **f** Stacked bar-charts of observed dPCR timeseries genome copy frequency summaries of edited T cells and K562 cells using SpCas9 with and without repair inhibitors/AAV-transduction 5 minutes to 14 days post editing. Data are shown as mean ± s.d. **g** ODE fitted curves (lines) modelled using the first 24 h of CLEAR-time dPCR data on T cells (left) and K562 cells (right) edited cells. Dots and error bars indicate mean ± s.d. of observed dPCR data. **h** Comparison of the average half-life of DSBs generated from uncleaved sequences and resolved as a repair product across all treatments in T cells (left) and K562 cells (right). **i** Pie-charts of the likelihood of DSB resolution into each measured repair product or remaining unresolved per hour. Calculated using the ratio of rate coefficients and the sum of all repair outcome rate coefficients. **j** Average number of DSBs necessary to generate each repair product. All data represents n = 3 technical replicates unless stated otherwise. All data derived from modelling (n = 3 replicates, 1000 bootstraps per condition) is shown as mean ± s.d. Source data are provided as a Source Data file.

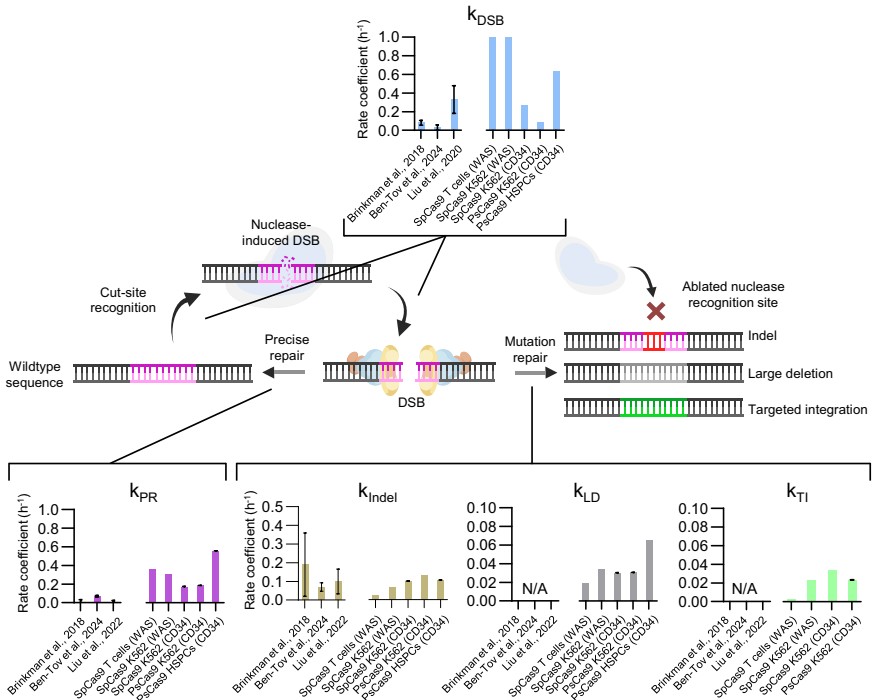

**Fig. 6 | Model of designer nuclease cleavage and repair.** Graphical representation of designer nuclease cleavage and cellular repair. Bar charts illustrate the rate coefficients calculated through the different phases of the DNA cleavage and repair process in comparisons with previously published articles addressing the DNA repair kinetics with Sanger sequencing + amplicon deep sequencing + LM-PCR (Brinkman et al.), LM-PCR (Ben-Tov et al.), and dPCR + amplicon deep sequencing (Liu et al.) vs. CLEAR-time dPCR (this manuscript). CLEAR-time dPCR data are shown as mean ± upper/lower limits of 95% CI (n = 3 replicates, 1000 bootstraps per condition) linked to Figs. 4 and 5. Data points from published literature shown as mean ± s.d. Source data are provided as a Source Data file.

enriches our understanding of these dynamics by assessing other repair products but also uncovers elusive repair activities. In some cases, the frequency of precise repair is nearly 40-fold greater than previously thought.

## Discussion

This study presents CLEAR-time dPCR, a comprehensive, versatile, and modular digital PCR technique that provides a thorough overview of genome integrity at a targeted site following the use of DSB-inducing editor tools. Although dPCR has been implemented to quantify aberrations such as indels, aneuploidy, and large structural variants[14,67,68], they neglect to do so in an absolute or comprehensive manner, resulting in a limited overview of the information that could be acquired. We addressed this by developing synergistic mutation-detecting assays, including a set of reference assays and a double-

normalization strategy to capture information that would otherwise be unaccounted for.

These features allowed us to reveal previously unreported significant biological information with unprecedented precision, including:

1) Precise (error-free) DNA repair mediated by NHEJ
2) Recurrent designer nuclease cleavage events
3) Kinetics of DNA repair outcomes in primary cells
4) Timing of DSB resolution
5) Designer Nuclease activity

Furthermore, we demonstrate how CLEAR-time dPCR enabled us to rapidly track and quantify various mutations in a temporal fashion.

In our comparative analysis, we evaluated amplicon deep sequencing, non-reference normalised dPCR, long-read sequencing,

Sanger sequencing (TIDE/ICE), flow cytometry, CAST-seq and illustrated the specific characteristics of each and found CLEAR-time dPCR to reliably combine the information derived from orthogonal methods into a single sensitive (~2% sequence detection capacity within the larger population) yet broad procedure.

In contrast to conventional dPCR assays, which provide a specific set of information, the synergy of CLEAR-time dPCR's approach produces a more comprehensive and revealing picture of complex genomic events, surpassing the sum of their individual contributions. Using CLEAR-time dPCR, we simultaneously measured the frequencies of indels, large deletions, DSBs, targeted gene integration, and other chromosomal aberrations amongst wildtype sequences in clinically relevant cell types across a broad range of therapeutic gene targets without a pre-processing step for the edited cells or need for high molecular weight DNA extraction.

Multiplexing the mutation detection assays with respect to a reference enables the absolute quantification of mutated sequence as a proportion of the total pool of loci[69]. This capability significantly simplifies the process, which would typically require a multitude of laborious techniques to obtain a comprehensive overview of the modifications induced by designer nucleases at targeted (and off-target) loci. Applying the technique in this manner allowed us, in some circumstances, to detect up to 92% of total loci with aberrations that would otherwise be unaccounted for using amplicon sequencing methods.

The outcome of designer nuclease-edited cells is a heterogeneous population of mutated sequences, which is likely attributed to the simultaneous engagement of multiple repair pathways associated with varying degrees of error and the resulting cleavage patterns by nucleases[39,70,71]. We edited a broad range of genomic targets and found a high degree of variability regarding the frequency of mutated sequences and editing efficiency. As there was seemingly no correlation between the frequency of DSBs quantified at early timepoints and the number of final indels, this may be attributed to chromatin accessibility for Cas9 or repair pathway activation, as seen previously[72]. This opens avenues for further studies to explore the relationships between chromatin state, DNA sequences, and repair mechanisms.

Furthermore, we propose CLEAR-time dPCR as a valuable tool for longitudinal in vivo studies for its capability to provide snapshots of the mutational status of the edited loci, utilising any possible variation in the mutation distribution as a proxy to track transplant clonogenicity and carcinogenesis. Not only did we show substantial loci variability between the edited HSPC treatment groups, but more importantly, between the initially edited cells and those after transplantation. As previously reported, cells with targeted integrations are dramatically reduced[73]. Additionally, the significant variation in mutation distribution among mice within the same group, as well as compared to the pre-transplant population, suggests a reduced number of progenitor cells. This variation also indicates that treated stem and progenitor cells carrying small indels or wildtype alleles have a higher chance of survival.

We therefore highlight a need for CLEAR-time dPCR in post-clinical follow-ups in patients who have received targeted cell therapies to detect and monitor the presence of both augmented therapeutically beneficial cells and, more importantly, cells harbouring potentially genotoxic aberrations.

Although unwarranted cellular repair activity constraints exist, nuclease-based targeted genome editing remains attractive due to its relative ease and curative potential. The NHEJ and HDR repair pathways exist in a state of competition that favours NHEJ due to its persistence through every cell cycle stage[74,75]. However, this can be tipped in favour of HDR when NHEJ is suppressed and thus prove beneficial for HDR-mediated gene editing[76]. Thus, selective and potent repair-inhibiting properties of cancer therapeutic compounds have found a potential role in gene therapy to improve HDR efficiency[77–85]. In our

attempt to define these compounds in a gene therapy context, we demonstrated the biases with relative quantitative methods like amplicon sequencing, and how double-strand breaks, and on-target deletions obscure the reality of genomic status. In comparison to methods described in previous studies[43,49,86], our CLEAR-time dPCR method stands out in its ability to comprehensively quantify various genomic aberrations while effectively minimising biases inherent in these techniques, particularly those related to large deletions, DSBs, and chromosomal aberrations. This methodology provides enhanced reliability in measuring DNA repair dynamics and assessing the performance of designer editors and therapeutic interventions across diverse research settings. DNA repair inhibitors delay the resolution of DSBs and cause evaluation biases when studying designer nuclease activity and HDR outcomes. In corroboration with previous work, we observed increased large deletions, which is likely due to redundant DNA-PKc-independent NHEJ pathways facilitating error-prone repair[47], however, we found that the presence of a donor template helps attenuate this phenomenon and appears to be cell type dependent[16,49]. Additionally, in multiple instances, over time, alleles with large deletions were lost from the total pool of loci, suggesting a survival disadvantage or other negative feedback mechanisms correlated with extensive end-trimmed repair.

Cullot et al.[49] also report significantly higher rates of large deletions when using AZD7648, but their analysis at 3 days post-electroporation, without removal of DSB repair-inhibitors, introduces bias that potentially misrepresents 50–80% of affected loci. This bias aligns with their HDR data discrepancies, underscoring the caution needed in interpreting these results. Thus, we encourage a more measured interpretation to avoid overstating the impact of AZD7648 or other DNA repair inhibitors in gene editing.

TIEs, in combination with CLEAR-time dPCR, have shown many useful basic and translational biological features previously only hypothesised. These experimental observations regarding the extent of cleavage and associated mutations were further investigated and corroborated through kinetics and modelling analysis. Time-course experiments were designed to track loci status at various time points, thereby measuring nuclease and repair activities and revealing their kinetics. For instance, the trafficking time of RNPs post-electroporation was identified, showing a lag of 10-30 minutes before the first cleavage and a peak activity occurring 4-8 h afterward. Additionally, therapeutically relevant cells were analysed for their response to DNA damage, showing that error-prone repairs (indels, large deletions) and HDR occur with a delay compared to NHEJ's precise repair. This indicates a shift in cellular responses to recurrent cleavages, as previously documented with single-cell imaging in cell lines[41], and emphasises that mutation generation may rely on a series of previous precise repair events.

Evaluating the precision of NHEJ to correctly repair a DSB is challenging, primarily due to the inability to differentiate wildtype and precisely repaired sequences. As we began to understand these various activities, we found that mathematical modeling using ODEs better elucidated the complex dynamics between nuclease-induced DSBs and cellular repair responses. Building on previous ODE models, our model tracks multiple mutation products, including small indels, and accounts for cellular repair activities with and without repair inhibition.

As previous methods to delineate accurate NHEJ-mediated repair are biased by a lack of a comprehensive aberration assessment, they fail to capture the full extent of precise repair and recurrent cleavage events. The inhibition of NHEJ provided experimental evidence supporting its crucial role in precise DNA repair without mutations. Our model quantified these dynamics, revealing that DSB repair can occur within minutes of a cut, highlighting the remarkable speed at which cellular repair mechanisms operate, with a DSB half-life measured at approximately 1-2 h, with minimal variation between primary and

immortalised cells. With precise repair accounted for, we were able to calculate the frequency of DNA cleavage by the nuclease and thus describe the general biological dynamics that determine the likelihood of inducing specific mutations at a locus. As hypothesised, most DSBs are resolved with precise repair, which occurred regardless of gene target, cell type, or if repair was largely inhibited with TIEs. Although HDR was still active, this likely does not fully account for the relatively high number of precise repairs seen in repair-inhibited cells and is more likely due to incomplete repair inhibition, activity of alternative yet high-fidelity repair mechanisms, or a combination thereof. We measured that in Cas9 edited human primary cells, >2.5 cuts are required to induce an indel, whereas more than >10 cuts are necessary for a large deletion. Interestingly, staggered-end cleavages resulted in similar mutation-prone repairs compared to blunt-end cleavages in a side-by-side testing.

In summary, our kinetics experiments combined with the CLEAR-time dPCR evaluation provide enhanced clarity on genomic stability in response to designer nuclease cleavage, both with and without the application of TIEs. Although attempts have been made to avoid DSB induction using Cas nickase-based editors, recent findings indicate that DSBs, along with indels and gross chromosomal aberrations, are still produced[52-54]. With the capability for single base discrimination, CLEAR-time dPCR can be adapted to quickly and accurately assess aberrations from emerging DNA editors, thereby improving their specificity and tracking their activity.

Despite the different advantages, CLEAR-time dPCR is primarily a comprehensive method to analyse a set of mutations on known sites, not a de-novo off-target discovery technique. Therefore, to leverage the modular capabilities of CLEAR-time dPCR, one should derive prior knowledge of off-target sites with genome-wide methods. The use of two-colour detection in our assays reduces technical variability across samples and costs[87]. However, this can be improved by taking advantage of new multi-colour platforms, which will allow for combining multiple mutation-detecting assays as well as facilitating internal normalisation to further reduce variability. CLEAR-time dPCR is a PCR-based technique, and therefore sequence-dependent; regions with limited base diversity may pose challenges when designing assays. CLEAR-time dPCR's key feature of multiplexing assays necessitates the validation of each assay and works best with consistent annealing temperatures across all primers and probes. The sequence of interest can therefore impose challenges, especially in low diversity regions, when designing assays, although this can be tackled using locked oligonucleotides[88]. Additionally, donor templates with long homologous sequences (i.e., >60 bp) can interfere with the flanking assay because the assay is unable to differentiate them. We address this issue by designing the flanking assays further away from the cleavage point and/or using a shorter ssODN template by placing the flanking assay outside the homology arms, though we acknowledge that this may not be feasible for all gene editing strategies.

In conclusion, CLEAR-time dPCR fills a critical niche in the post-editing analysis toolbox by offering a rapid, comprehensive, and specific overview of genome integrity through the characterisation of unintended aberrations and mutations resulting from contemporary gene-editing techniques. Implementing CLEAR-time dPCR to derive richer insights into designer nuclease activity will enhance the evaluation of safety profiles, activity characterisation, and the development of strategies for novel gene editing methods, facilitating their effective translation into clinical applications. Furthermore, for the first time to our knowledge, CLEAR-time dPCR enabled the unbiased measurement and the validation of the elusive 'precise repair' as the primary DNA repair mechanism, thereby confirming the cycles of repair and re-cutting by designer nucleases. Finally, we introduced a relatively simple yet powerful method to study DNA repair mechanisms, revealing fundamental insights and paving the way for extensive mechanistic studies on designer editor activity.

## Methods

A detailed information list regarding primers, sgRNA, donor sequences, PCR conditions and material are summarised in Supplementary data 1-5.

### Cell culture

**HSPCs.** CD34+ HSPCs were isolated by positive selection from conditioned peripheral blood from healthy male donors using the human CD34 MicroBead Kit (Miltenyi Bioscience) following the manufacturer's instructions. Viability and purity were assessed with flow cytometry. Cells were cultured at $10^6$ cells/ml at 37 °C, 5% CO2 in StemSpan SFEM Haematopoietic Cell Culture Medium (Stemcell Technologies), supplemented with Recombinant Human TPO (100 ng/ml; Peprotech), Recombinant Human SCF (100 ng/ml; Peprotech), Recombinant Human Flt3-Ligand (100 ng/ml; Peprotech), Recombinant Human IL-3 (100 ng/ml; Peprotech), and Penicillin-Streptomycin (1%; Gibco). Cell cultures were passaged every 2–3 days.

**iPSCs.** J1 iPSC cells (Synthego) were cultured in Cellartis DEF-CS 500 COAT-1 coated plates and maintained in Cellartis DEF-CS 500 Basal Medium supplemented with Cellartis DEF-CS 500 Additives (DEF-CS GF-1, GF-2, and GF-3). Reagents for iPSC culture were purchased from Takara Bio. Cells were maintained at 37 °C in a 5% $CO_2$ atmosphere.

**PBMC derived T cells.** Peripheral blood derived PBMCs were cultured at $10^6$/ml at 37 °C, 5% CO2 in TexMACS Medium (Miltenyi Bioscience) supplemented with Human serum (5%; Merck), Recombinant Human IL-2 (50 ng/ml; Peprotech), and Penicillin-Streptomycin (1%; Gibco). To activate and expand T cells, PBMCs were cultured overnight to exclude plastic adherent monocytes and activated the following day using Human T-Activator CD3/CD28 Dynabeads (1:1 bead/cell ratio; Gibco) and left to activate for 72 h. Dynabeads were then removed following the manufacturer's instructions and allowed to expand for 72 h prior to editing.

**K562 cells.** K562 cells (ATCC CRL-243) were cultured at $10^6$/ml at 37ºC, 5% $CO_2$ in RPMI 1640 Medium, GlutaMAX (Gibco), supplemented with FBS (10%; Gibco), and Penicillin-Streptomycin (1%; Gibco), and passaged every 2-3 days.

**AAVpro 293 T cells.** AAVpro® 293 T Cell Line (Takara # 632273)cells were cultured at density of $6 \times 10^4$ cells/cm², 5% $CO_2$ in DMEM, high glucose, GlutaMAX supplement (ThermoFisher), supplemented with HEPES (25 mM; ThermoFisher), FBS (10%), and Penicillin-Streptomycin (1%). Cells were passaged every 3-4 days, washed with DPBS (ThermoFisher), and detached using Trypsin-EDTA (0.05%; ThermoFisher).

### PsCas9 production and purification

For kinetics experiments targeting *CD34*, we used an engineered version of PsCas9[64], ePsCas9, from now on referred to as 'PsCas9'. PsCas9 protein was produced in E. coli BL21 λDE3 Star using the T7 expression system, and all plasmids were based on pET24a. Freshly transformed cells were grown overnight in LB medium before inoculating with 800 ml of TB medium the following day. Expression cultures were then grown at 37 °C with vigorous shaking until 0D600 ~ 2. The temperature was then lowered to 18 °C, and IPTG was added after 1 h to a final concentration of 200 μM. The cells were then left at 18 °C overnight and harvested by centrifugation the following day. Cells were resuspended in 20 mM HEPES pH 7.5, 150 mM KCl, 5% glycerol, and 1 mM DTT. Lysis was performed by high-pressure disintegration and clarified by centrifugation at $15,000 \times g$ for 30 min. The supernatant was then applied to a 5 ml HisTrap column (Cytiva) and washed with a buffer consisting of: 20 mM HEPES pH 7.5, 150 mM KCl, 5% glycerol, 20 mM imidazole, and 1 mM DTT. Bound proteins were then eluted by washing with the same buffer supplemented with 300 mM imidazole. Protein-

containing fractions were then applied to a Superdex 200 10/600 column (Cytiva), equilibrated with 20 mM HEPES pH 7.5, 300 mM NaCl, 5% glycerol, and 1 mM DTT. The protein peak was then collected, concentrated to 10 mg/ml, flash frozen in small aliquots, and stored at −80 °C until use.

### sgRNA and ssODN design

sgRNAs targeting *WAS*, *BTK*, *IL7R*, and *SH2D1A* genes have been described previously[7,56,89,90]. The ssODN and sgRNAs targeting *CD34* for SpCas9 and PsCas9 have been described previously[16]. Chemically modified synthetic sgRNAs and ssODNs harbouring 2'-O-methyl analogues and 3'-phosphorothioate non-hydrolysable linkages at the first three 5' and 3' nucleotides were purchased from IDT. Primers, sgRNA and donor sequences are summarised in Supplementary data 1-3.

### AAV production

AAV transgene plasmids encoding GFP with homology arms targeting WAS and BTK were generated as described previously[7,90]. A total of $7.5 \times 10^8$ AAVpro 293 T cells were seeded in five 15-cm cell culture dish ($1.5 \times 10^7$/dish). Medium was replenished 2 h prior transfection. Cells were transfected at a 1:1:2 molar ratio of pTransgene:pAd5:pRep2Ca-pAAV6, with a 2:1 molar ratio of polyethylenimine (Linear, MW 25000; Polysciences): DNA. Prior to collection, transfection efficiency using GFP expression was assessed 72 h post-transfection using flow cytometry. Both cells and supernatant were harvested and 40% PEG8000 (polyethylene glycol 8000; VWR) was added to the supernatant and incubated for 3 h at 4 ˚C to precipitate AAVs whereas the cell pellet was lysed with 3 freeze-thaw cycles. Both PEG pellet and cell lysate were combined and incubated for one hour at 37 ˚C with benzonase nuclease (200U/ml; Sigma-Aldrich) and 30 minutes with 0.5% deoxycholic acid (Sigma-Aldrich). The lysate was then subjected to an iodixanol gradient ultracentrifugation to purify AAVs, concentrated using a 100 K Amicon Ultra-15 Centrifuge Filter (Sigma-Aldrich), and the AAV titre quantified by droplet digital PCR targeting ITR regions and GFP. AAVs were aliquoted and stored at −80 °C until use. See Supplementary Data 3 for donor template sequences.

### Genome editing

SpCas9 or AsCas12a (30 pmol; Integrated DNA Technologies) was complexed with a single guide RNA (225 pmol; Synthego, CRISPR evolution sgRNA EZ Kit) to form a ribonucleoprotein complex (RNP) for 10 minutes at room temperature. SpCas9 (SpCas9-V3 IDT) or PsCas9 at 4.5 μM in the final transfection reaction, and sgRNAs at 1:6.7 Cas9: sgRNA molar ratio were assembled to RNP complexes at room temperature. See Supplementary Data 2 in for the full list of sgRNA sequences used. HSPCs were transfected with ssODNs at 10 μM in the final transfection reaction and/or RNP complexes. $1.5 \times 10^6$ cells were resuspended in 20 μl P3 (Primary cells; Lonza) or SF (Cell lines; Lonza) nucleofector solution, mixed with the complexed RNPs and electroporated using the 4D-Nucleofector system (Lonza). Using 16-well Nucleocuvette Strips, the CA-137, EO-115, and FF-120 programs were used to electroporate HSPCs, T Cells, or K562 cells, respectively. To prepare RNP complexes for iPSCs, SpCas9 (SpCas9-V3 IDT) and sgRNA were mixed in a 1:1 molar ratio and incubated for 10 min at room temperature. The RNPs were kept on ice until electroporation. For each reaction, $0.5 \times 10^5$ cells were combined with 40 pmol SpCas9, 40 pmol sgRNA, 40 pmol ssODN and, 40 pmol electroporation enhancer oligo. In reactions without ssODN, 80 pmol of the electroporation enhancer was used. Prior to electroporation, iPSCs were washed with PBS, the cell number and viability were assessed using a Vi-cell XR (Beckman Coulter) cell counter, and the cells were resuspended in nucleofection buffer (P3 Primary Cell 96-well Nucleofector™ Kit, Lonza). Immediately before transfections, ssDNA donor, electroporation enhancer (IDT), and cells were added to the RNPs. The electroporation mix was then transferred to the nucleocuvette strip, and

the Lonza 4D-Nucleofector program CA-137 was used for electroporation. For experiments using larger cell numbers ($10^7$ cells), the 100 μL Nucleocuvette Vessel was implemented and the volume of Nucleofector solution and RNP was scaled accordingly. After electroporation, cells were transferred to 6- or 12-well cell culture plates containing pre-warmed medium with DMSO or the small molecule inhibitors AZD7648 and ART558, with a final concentration of 0.5 μM and 3 μM, respectively. Unedited cells, whereby no RNP were electroporated but otherwise followed the same process, were used as controls for dPCR and sequencing analysis. Cells were transduced with AAVs at an MOI of 12,500 unless stated otherwise.

### Small molecule compounds and treatment

DNA-PK inhibitor AZD7648 and POLQ inhibitor ART558 were purchased from MedChemExpress or Cambridge Bioscience. POLQi2 (WO2020243459) was provided by AstraZeneca (Gothenburg, SE). All compounds were dissolved in dimethyl sulfoxide (DMSO) at a stock concentration of 10 mM. Transfected cells were cultured in medium containing inhibitors from 0 to 24 h post-transfection, followed by a cell wash and culture in control medium.

### Genomic DNA extraction

For dPCR experiments, to preserve DNA integrity, High Molecular Weight (HMW) DNA extractions were performed using the Monarch® HMW DNA Extraction Kit for Cells & Blood (NEB) following the manufacturer's instructions. 20 μl DNA was digested with 0.5 μl DraI and 1.55 μl rCutsmart buffer (NEB) at 37 °C for 20 minutes. For VCN and concatemer analysis, 1 μl DNA from AAV transduced cells was digested with 1 μl BSU36I and 2 μl rCutsmart buffer (NEB) at 37 °C for 20 minutes and diluted 1/10 with $H_2O$. DNA concentration was quantified using a Nanodrop™ OneC UV-Vis Spectrophotometer. For CAST-Seq and Sanger sequencing, DNA extractions were carried out using Monarch Genomic DNA Purification Kit (NEB) following the manufacturer's instructions. For kinetics experiments targeting *CD34*, cells were immediately snap frozen at indicated time points.

### Indel quantification

DNA extracted from cells 3 days post-nucleofection were PCR amplified with sequencing primers (See Supplementary Data 1 for a complete list of primers used) and purified using Monarch PCR & DNA Clean-up Kit (NEB). For Sanger sequencing, samples were processed using the Eurofins Genomics TubeSeq NightXpress service with 2 μl forward primer pre-mixed with 15 μl PCR-purified amplicons (10 ng/μl). Indel frequency assessment was performed using Inference of CRISPR Editing (ICE) analysis (Synthego Performance Analysis, ICE Analysis. 2019. v3.0. Synthego) with a model co-efficient of determination $R^2 > 0.9$. For the T7EI heteroduplex assay, PCR-amplified unedited controls and edited samples underwent heteroduplex formation by denaturing for 5 minutes at 95 °C and then slowly reannealing (2 °C per second between 95 °C and 85 °C, followed by 0.1 °C per second from 82 °C to 25 °C). Heteroduplexed amplicons were then digested at 37 °C with EnGen T7 Endonuclease 1 (T7E1, NEB) for 20 minutes. Electrophoresed samples were analysed by ImageJ (Version 1.54 g) to calculate the relative percentage of indels[91].

### Next generation sequencing

*CD34* amplicons were generated with Phusion Flash High-Fidelity 2x Mastermix (F548, Thermo Scientific) in a 15 μL total reaction, containing 1.5 μL of genomic DNA extract and 0.25 μM each of primers (See Supplementary Data 1 for the list of primers used). All amplicons were purified using HighPrep PCR Clean-up System (MagBio Genomics). Size, purity, and concentration of amplicons were determined using a fragment analyzer (Agilent). To add Illumina indexes to the amplicons, samples were subjected to a second round of PCR. Indexing PCR was performed using KAPA HiFi HotStart Ready Mix (Roche),

0.067 ng of PCR template, and 0.5 μM of indexed primers in the total reaction volume of 25 μL. Samples were purified with the HighPrep PCR Clean-up System (MagBio Genomics) and analysed using a fragment analyzer (Agilent). Samples were quantified using a Qubit 4 Fluorometer (Life Technologies) and subjected to sequencing using the Illumina NextSeq system according to the manufacturer's instructions. Demultiplexing of the *CD34* RIMA NGS data was performed using bcl2fastq software. Sequences were analysed with RIMA[92] and CRISPResso2[93] with a lower limit of acceptance >5000 reads per target.

### Large deletion sequencing
Amplicons approximately 5000 bp in length were generated by PCR reaction with primers flanking ~2500 bp of the cleavage site and purified using AMPure XP Bead-Based Reagent (BECKMAN COULTER). 300 ng of amplicons were fragmented, end-repaired, and dA-tailed using NEBNext Ultra II FS DNA Library Prep Kit (NEB). The fragments were then ligated with adapters from the DNA Library Prep Kit, following the manufacturer's guidelines. Libraries were purified with AMPure XP Bead-Based Reagent and barcoded using NEBNext Multiplex Oligos for Illumina (Dual Index Primer Pairs, NEB) for NGS. Finally, double-size selected purification was performed with AMPure XP Bead-Based Reagent to obtain 200–600 bp-length amplicons. Briefly, the purified amplicons were incubated with 0.5 times the total volume of magnetic beads for 5 min, magnetically separated and the harvested supernatant incubated with 0.2 times the volume of beads, again for 5 min. The supernatant was removed, and the purified amplicons remained, followed by two 80% ethanol washes, and eluted in TE buffer. The amplicon libraries were then quantified using droplet digital PCR with the "ddPCR Library Quantification Kit for Illumina TruSeq" (Bio-Rad). The denatured library (10–20 pM) was supplemented with 20% PhiX Control v3 (Illumina) and sequenced using MiSeq (Illumina) with reagent Kit v3 (600 cycles). Large deletion data were analysed using Galaxy (version 24.0.rc1).

### Digital PCR
In triplicate for each assay, DraI (NEB) digested DNA (25 ng per assay) was mixed with 2x digital PCR supermix for probes (No dUTP) (Bio-Rad), vortexed, and distributed with equal volumes into a ddPCR 96-Well Plates (Bio-Rad). Individual assay mixtures composed of primers (0.8 μM) and probes (0.25 μM) were then added to the ddPCR 96-Well Plates containing the DNA/supermix mixture. No-template controls, whereby no DNA was assayed, were included as a control for each CLEAR-time dPCR assay that was run. Droplets were generated using Automated Droplet Generation Oil for Probes (Bio-Rad) and DG32™ Automated Droplet Generator Cartridges with the automatic droplet generator (Bio-Rad QX200 AutoDG), and the plate was sealed with a PCR Plate Heat Seal (Bio-Rad) and PX1™ PCR Plate Sealer (Bio-Rad). PCR was performed using the C1000 Touch™ Thermal Cycler with 96-Deep Well Reaction (Bio-Rad) and droplets read using a QX200 Droplet Reader (Bio-Rad). Droplet thresholds were manually set using QX Manager Software (Bio-Rad, Version 2.1) and processed in Excel (Microsoft). See Supplementary Data 1 and 4 for the full list of primer and probes used and PCR cycling conditions, respectively.

**dPCR Probes.** All probes used in this study were PrimeTime qPCR probes (IDT) with the 5' end fluorescent dye modification (6-Carboxyfluorescein "FAM", or Hexachlorofluorescein "HEX", 3' end quencher (Iowa Black® FQ "IBFQ"), and "ZEN" internal quencher. All probes were designed have a melting temperature 5-9°C greater than the primers. The Edge cleavage probe was oriented so the 5' end of the oligo is placed over the cleavage site to minimize non-specific binding in the case of small indels, i.e., +1 or −1. In the case of non-specific binding, we utilized Affinity Plus locked nucleotide (LNA) oligos (Integrated DNA technologies) and locked the nucleotides on either side of the prospective cleavage site.

### Limit of detection
**Cassette generation.** Cassettes were created with three PCR reactions. The first PCR generated amplicons of the wildtype, a simulated large deletion at the *CCR5* locus, and also a reference sequence, each with the addition of flag sequences. A second fusion PCR was carried out to create the full-length amplicons of each cassette fused to the reference amplicon. A final nested PCR was performed using primers downstream of the initial PCR, and cassettes were purified using the Monarch PCR & DNA Cleanup Kit (NEB) following the manufacturer's protocol. See Supplementary Data 1 for a complete list of primers. Gel electrophoresis of the library DNA was performed, and the desired DNA was extracted from the gel, followed by cleaning up with the Monarch DNA Gel Extraction Kit (NEB). The concentration of wild-type *CCR5* and unbalanced large deletion cassettes was quantified using dPCR. A serial dilution of decreasing *CCR5* large deletion concentrations (100%, 45%, 8%, 4%, 2%, 0.8%, 0.4%, and 0% of the large deletion cassettes) was performed using samples of large deletion cassettes within a *CCR5* wildtype population to create a known concentration gradient. Detection sensitivity was quantified as described previously[49]. Briefly, to calculate the limit of blank (LoB), dPCR derived copies from 20 replicates of the 3' sequence of the wildtype cassette was normalised to the reference sequence and again to the average of the replicates and added to the variance multiplied by the 95% confidence interval.

$$\text{LoB} = \mu_{WT;3'} + 1.645(\sigma_{WT;3'})$$

Where;

$\mu_{WT;3'}$ = Mean reference normalised 3' sequence
1.645 = 95% confidence interval
$\sigma_{WT;3'}$ = Standard deviation of reference normalised 3' sequence.
The limit of detection (LoD) was calculated by normalizing the large deletion cassettes to the reference sequence amongst the wild-type sequences in the serial dilutions and applying dilution factors. The LoD was calculated using the variance of the lowest normalised/dilution factor corrected sample in the dilution series detected above the LoB.

$$\text{LoD} = \text{LoB} + 1.645(\sigma_{LD;low})$$

Where

LoB = Limit of Blank
1.645 = 95% confidence interval
$\sigma_{LD;low}$ = Lowest concentration test sample above LoB.

### Human CD45-positive cell isolation
Intron 1 of the gene encoding XIAP in human female peripheral blood-derived HSPCs was edited with high-fidelity Cas9 and transduced with AAV-6-encoding codon-optimised *XIAP* (*coXIAP*) and *GFP* or *GFP* alone flanked with homology arms to target the *XIAP* locus. Unedited, RNP-only, and RNP + AAV-transduced HSPCs were transplanted into four female immunodeficient NSG mice (5×10^5 cells per mouse; 8-week-old) per treatment group and left for 16 weeks. Femoral and tibial bone marrow were flushed with MACS buffer and washed again with MACS buffer. Human CD45-positive cells were positively enriched using hCD45 microbeads (Miltenyi Biotec) and MS columns (Miltenyi Biotec) following manufacturer instructions. Isolated cells were checked for GFP expression using flow cytometry, and the remaining were used for DNA extractions.

### Kinetics
**Genome editing, inhibitor treatment, and AAV transduction.** Cells were electroporated and split into separate treatment groups (RNP-only, RNP + repair inhibitors ART558 (3 μM) & AZD7648 (0.5 μM), RNP + AAV (MOI; 12,500), and RNP + AAV (MOI; 12,500) + repair

inhibitors ART558 (3 µM) & AZD7648 (0.5 µM). DNA extractions were performed using the Monarch Genomic Purification Kit (NEB) at multiple timepoints ranging from 1 minute to 14 days post-nucleofection and analysed using CLEAR-time dPCR. A complete medium change was performed for each treatment group 24-h post-nucleofection, without repair inhibitors or AAVs.

**Kinetics curve fitting and analysis.** Curve fitting was performed using a set of ODEs to describe the three-state model of Cas9-induced DNA cleavage and subsequent cellular repair (Supplementary Information 1; Eq. 19). These ODEs continuously capture the rate changes of each measured loci population over time based on corresponding fixed rate coefficients. To determine the best estimates for these parameters, the model generates 1000 bootstraps from the replicate data and minimises the residuals between the observed data and the fitted model. We followed the established assumptions that Cas9 cutting and repair individually follow first-order kinetics and RNP activity is delayed by cellular trafficking and incorporated this into model fitting[42]. Based on the assumption that TIEs do not directly alter Cas9 cleavage activity, the 95% confidence interval for the $k_{dsb}$ value were determined based on simultaneous fitting to RNP data with and without DSB repair inhibitors. To account for the delay in Cas9 cleavage activity due to RNP trafficking into the nucleus following electroporation, $k_{dsb}$ was multiplied by D(t), and is dependent on the 'delay' parameter also determined by the fitted data.

The equations regarding DSB generation (Eq. 20), DSB resolution (Eq. 21), repair likelihood (Eq. 22), and DSBs per repair (Eq. 23) derived from the rate coefficients are reported in Supplementary Information 1.

**CAST-Seq**
CAST-Seq was performed as described previously[14]. We randomly fragmented 500 ng of genomic DNA using the NEBNext Ultra II FS DNA Library Prep Kit (NEB) to achieve an average size of 350 bp. Linker oligos were ligated to both ends of fragments, followed by an initial PCR utilising an on-target bait primer, linker prey primer, and decoy primer to prevent amplification of on-target sequences. A second PCR was performed with nested primers and adapters to the amplified sequences. Samples were barcoded using NEBNext Multiplex Oligos for Illumina for NGS in a third PCR reaction, followed by PCR purification using AMPure XP Bead-Based Reagent (BECKMAN COULTER). Individual barcoded samples and the final pooled library were quantified by droplet digital PCR. The denatured library (10–20 pM) was sequenced using MiSeq (Illumina) with reagent Kit v3 (600 cycles) and supplemented with 20% PhiX Control v3 (Illumina). See Supplementary Data 1 for a complete list of primers. CAST-Seq data were analysed as previously described[28].

**Flow cytometry**
Cells were stained with 5 nM TO-PRO-3 (Thermo Fisher) or 0.1 µg/ml DAPI to quantify cell viability and GFP in AAV-transduced cells using an LSR II (BD Biosciences) flow cytometer. Flow cytometry data was acquired using BD FACSDivaTM (BD Biodata) and analysed using FlowJoTM V10.9 (BD Biosciences).

**Statistics & reproducibility**
All digital PCR experiments were performed as three technical replicates (or as otherwise stated in the text). Statistical analysis and graphical illustrations were performed using Prism (GraphPad, version 8.0.1 (255)). Statistical significance was defined when p-values were less than 0.05; all exact p-values are addressed within figure legends where possible. All statistical tests performed were two-tailed unless stated otherwise. No data were excluded from the analyses.

**Ethics statement**
NSG mouse experiments were carried out in accordance with United Kingdom Home Office regulations and received approval from the University College London Animal Welfare and Ethical Review Body (Project license 70/8241).

Informed written consent was obtained from healthy donors for the use of human CD34+ haematopoietic stem and progenitor cells (HSPCs), in accordance with the Declaration of Helsinki. Ethical approval was granted by the Great Ormond Street Hospital for Children NHS Foundation Trust and the Institute of Child Health Research Ethics Committee (08/H0713/87).

AstraZeneca has a governance framework and processes in place to ensure that commercial sources have appropriate patient consent and ethical approval in place for the collection of samples for research purposes, including use by for-profit companies.

**Reporting summary**
Further information on research design is available in the Nature Portfolio Reporting Summary linked to this article.

## Data availability
Source data are provided with this paper. The CLEAR-time dPCR analysis files are available in the source data files. Next generation sequencing data generated by this study are publicly available in the NCBI SRA under the BioProject accession number PRJNA1332804 https://www.ncbi.nlm.nih.gov/sra/PRJNA1332804. Source data are provided with this paper.

## Code availability
Kinetics modelling pipelines are available at github.com/A-Chalk/DSB-Kinetics-Calculator under the GNU Affero general public license v3.0 (AGPL-3.0).

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

## Acknowledgements

We would like to thank Dr. Ayad Eddaoudi and his team (Flow Cytometry Core Facility, University College London) for assistance with flow cytometry; Dr Mark Kristiansen and his team (UCL Genomics Core Facility, University College London) for assistance with NGS sequencing; Dr. Giovanni Di Veroli (AstraZeneca, Cambridge) for his advice regarding kinetics modelling; Dr. Ben C. Houghton for providing TALEN edited samples and reagents and finally Georges Blattner for the CLEAR-time dPCR proof of concept. The work was supported by the Wellcome Trust, United Kingdom (217112/Z/19/Z). This project has received funding from the Innovative Medicines Initiative 2 Joint Undertaking (JU) under grant agreement No 945473 (ARDAT). The JU receives support from the European Union's Horizon 2020 research and innovation programme and EFPIA.

## Author contributions

N.W. designed and performed experiments, analysed, and illustrated data, and wrote the manuscript. J.A.C. performed kinetics experiments,

wrote the modelling scripts to analyse these datasets, and wrote the manuscript. Y.H., performed NGS experiments, analyzed data, and contributed to the manuscript. S.M.P., P.A., S.W., St.S., R.R., Sr.S., W.N.F., M.R. and C.R.J. performed experiments, analysed data, and reviewed the manuscript. S.C.S., A.C.A.M., performed murine experiments and reviewed the manuscript. G.S., R.N., C.B., G.S., A.C., M.H.P., M.M., and A.J.T. provided funding, supervised the experiments, and reviewed the manuscript. G.T. designed and supervised the study and the experiments, analysed data, provided fundings, and wrote the manuscript.

## Competing interests

N.W., A.C., A.J.T. and G.T. filed a patent application for the MEGA/CLEAR-time dPCR method (PCT/GB2022/052772). M.M., W.S., M.H.P. and Sr.S. filed patent applications on the drug inhibitors for enhancing HDR (WO2023052508A2; WO2023220418A3). A.J.T., C.B., A.C., G.T., and G.S. were also supported by the National Institute for Health and Care Research Biomedical Research Center at Great Ormond Street Hospital for Children, National Health Service Foundation Trust, and University College London. A.J.T. is on the Scientific Advisory Board of Orchard Therapeutics, Generation Bio, Carbon Biosciences, and 4BIO Capital. C. Booth has performed ad hoc consulting in the past 3 years for SOBI and Novartis and educational material production for SOBI and Chiesi. P.A., S.W., C.R.J., G.S., R.N., M.M. and G.T. are presently employed by AstraZeneca and may be AstraZeneca shareholders. The remaining authors declare no competing interests.
