## [Transparent Peer review file · Nature Communications]

Unveiling the Cut-and-Repair Cycle of Designer Nucleases in Human Stem and T Cells via CLEAR-time dPCR

Corresponding Author: Dr Giandomenico Turchiano

Version 0:

Reviewer comments:

Reviewer #1

(Remarks to the Author)

In this study, White et al. describe an interesting methodological framework, MEGA, for the detailed analysis of designer nuclease-mediated gene-editing outcomes. It is based on dPCR panels that provide comprehensive information including undesired repair outcomes (mostly large deletions and rearrangements) that are not normally detected in targeted sequencing approaches. MEGA lacks sequence-level resolution, but, combined with additional approaches, can become a widely used tool for assessing the performance of gene-editing strategies. From a mechanistic point of view, the study provides an interesting quantitative model that includes precise DSB repair and target re-cleavage, events that are predicted to occur, but have not been previously studied in such depth. I only have a few concerns and some minor points that the authors should address before publication of this work.

- The different strategies of MEGA and how each type of event is deduced from the results should be better explained. Understanding some of the approaches require substantial effort from the reader.

- I am not sure about what the authors refer to as "resection". In classical DSB repair terms, resected DSBs will still contain one intact DNA strand, and will therefore still be detected by dPCR (half of the signal, but the same number of positive droplets). Unless I am wrong, MEGA does not measure resection.

- The results of the long-term (14 days) experiments can be influenced by the selective advantage/disadvantage produced by the specific genetic modification. In agreement with this, the results of editing at different loci is variable (Figure 2c). This can put into question some of the conclusions such as the remaining activity of the nuclease after 72h and others. If authors want to draw conclusions from these long-term experiments, they should include MEGA analysis at a safe-harbor locus for comparison.

- The comparison of MEGA results with next-generation sequencing (Figure 3e) is somewhat unfair. The type of analysis performed on the NGS data is not sufficiently described. I believe that some types of analyses, e.g. analyzing coverage, may provide additional information perhaps comparable to MEGA.

Minor points:

- Throughout the manuscript. It is incorrect to consider DSBs an "allele".

- Line 25. "Recurrent nuclease cleavage events" is not a DNA repair mechanism.

- Line 112. The last sentence of the paragraph is not clear.

- Line 355. The target of the DNA repair inhibitors used should be indicated, and how they operate to increase targeted integration should be discussed.

- Line 362. The comparison of MEGA with sanger sequencing approaches is unfair and misleading, as the purpose of these approaches is to determine edited repair products and not to quantify DSBs. This should be indicated.

- Line 485. The average time for indel formation seems excessively long (6.9 h). Can this be influenced by the fact that many of these events occur after several rounds of accurate repair and re-cleavage? It may give the wrong impression that the repair events that result in indels takes this long.

- Line 545. MEGA does not provide information on repair mechanisms.

- Figure 1a. Is the scheme of large deletions correct? Shouldn't the chromosome be completely black.

(Remarks on code availability)

Reviewer #2

(Remarks to the Author)

The authors report the development of a modular dPCR assay for quantifying and tracking multiple genomic aberrations induced by CRISPR editing in multiple cell types, including HSPCs and T-cells. White, et al., conduct time course experiments to better understand CRISPR editing kinetics and fit their data to an ODE model to derive rate constants and DSB half-life. The authors propose that their modular dPCR assay would have clinical applications for tracking both desired and unintended editing outcomes.

Unfortunately, the authors fail to cite crucial references that seriously undermine the novelty of their work. It is unclear what is original about this work or what it adds to our understanding of CRISPR editing. The assay itself is simply multiple dPCR assays lumped together. Each of these assays on their own are not novel. Beyond a lack of originality, there is a question over the utility of the work. What are the advantages of using this modular dPCR assay, which would require design and optimization of amplicons for every on- and off-target of interest, vs. WGS or related methods (e.g. DISCOVER-seq, BLISS, etc). These NGS based methods could easily be combined with one or two existing dPCR approaches to provide much more information than is possible with MEGA.

There are further issues with insufficient validation, experimental design, and unsupported claims. However, the major issues of a lack of originality and unacceptable omissions of crucial references mean this work is not suitable for publication in Nature Communications.

Major concerns

1. The novelty of the work is questionable. Two of the primary claims to novelty are the modular dPCR assay and the analysis of kinetic analysis, but the authors insufficient discussion of the literature (see point 2 below) fails to describe relevant precedent. When considered in view of what has already been published, the current study appears to simply be the grouping together of a few non-novel ddPCR assays.
 - a. Together with long-read sequencing and other assays, ddPCR has been used to assess the frequency of large deletions, translocations, aneuploidy and other aberrations in the context of AZD7648 treatment (PMID 39604565).
 - b. An assay for nuclease induced DSBs has already been described (PMID 28737741, ref 22 in the manuscript). It is more robustly validated than the dPCR assay for DSBs reported in this paper.
 - c. The ddPCR assay in ref 22 has already been adapted for quantifying large deletions/chromosome arm level loss (PMID: 37794590).
 - d. Others have used ddPCR to quantify indels and knock-in efficiency.
 - e. Far more temporally precise examinations of CRISPR-editing have been published (PMIDs: 28737741 & 32527834).
2. Failure to properly cite and discuss relevant literature. The authors omit numerous crucial papers from their manuscript or fail to sufficiently or accurately relay the findings of these papers when discussed. These papers are perhaps the most salient publications to the authors' work—failure to cite and discuss them is unacceptable. As in point 1, it is unclear what this manuscript adds when considered in the context of the most relevant literature.
 - a. Cullot et al (PMID 39604565): This paper used ddPCR and other assays to examine CRISPR-induced large-scale genome aberrations promoted by AZD7648. This is the exact same DNA-PKcs inhibitor used by the authors.
 - b. Rose et al (PMID: 28737741). This paper reported the development of a ddPCR assay for absolute quantification of nuclease-induced DSBs, DSB-ddPCR. This paper characterized Cas9 editing kinetics using this assay and a rapidly inducible Cas9 variant, specifically DSBs and indels.
 - c. Liu et al (PMID 32527834). This paper reported a photo-activated CRISPR system and examined Cas9 editing kinetics with NGS, DSB-ddPCR, imaging and time-resolved ChIP. Notably, this paper made great advances in our understanding of cycles of error-free repair through the integration of their DSB-ddPCR, ChIP, and imaging experiments.
 - d. Tsuchida et al (PMID: 37794590): This paper utilized DSB-ddPCR to detect and quantify arm-level chromosome loss.
 - e. Failure to discuss genome-wide methods for detecting and quantifying DSBs, e.g. BLISS/BLESS/qDSB-seq (PMID 31127121).
3. Questionable utility. The report lacks novelty (see points 1 and 2 above), but the utility is also unclear. There are significant limitations (laborious design and validation of dPCR assays for every target and off-target of interest). Is it simply a matter of cost relative to WGS, DISCOVER-seq, etc? These genome-wide methods have a major advantage of being able to detect off-target edits, which MEGA-seq cannot do (unless the off-target is already known and a dPCR assay is designed

for it). WGS could also be easily combined with DSB-ddPCR or other ddPCR assay, a la Tsuchida et al, to detect chromosome arm level loss.

4. The authors state “The detection of indels demonstrated concordance across multiple techniques, including non reference normalised dPCR, sequencing based methods (NGS, TIDE, ICE), and the T7EI assay (Fig. 1g; Supplementary Fig. 1d). However, since these methods are inherently biased, amplifying primarily unbroken sequences with few or no base insertions or deletions, MEGA suggests that they underestimate indel frequency by approximately 17% in these samples.” There are major problems with these claims.

a. I do not see any NGS data in fig 1g or S1d. Targeted deep-sequencing with NGS is the gold-standard for quantifying indels and must be shown here.

b. To validate their assay, the authors must compare its performance against NGS at multiple sites, not just one.

c. The claim that the previously established assays are underestimating indel frequency must be backed up with evidence! Furthermore, dPCR also relies on amplification of the target sequence, so it is unclear what the authors’ argument is here.

5. The authors describe their approach as unbiased, but it is unclear what they mean by that. Authors need to clarify and validate this claim.

6. Insufficient validation of their dPCR assays, particularly for DSBs.

a. Needs to be compared to standard curves.

b. The authors assay needs to be benchmarked against DSB-ddPCR (see Rose et al, ref 22).

7. Need to characterize the tolerance of the dPCR probes to mismatches for indel detection. If they can tolerate 1 bp mismatches, their assay may undercount edits at some sites. The mismatch tolerance should be characterized at multiple sites.

8. In the first sentence of the manuscript, multiple newer versions of CRISPR editing approaches are referenced, including, prime editing, base editing, and PASTE. It is unclear if the MEGA approach has much utility in the context of these methods. NGS seems better suited to characterizing/quantifying the outcomes from these approaches. Though dPCR is well-suited for quantifying large-targeted integrations, this approach is not novel and it cannot confirm the sequence.

a. The location of the nicking guide would appear to complicate using MEGA with Prime editing. This nicking guide means 2 DSBs separated by dozens of nucleotides can be generated. It is unclear if the authors linkage assay would still work in this case.

b. If Prime editing, base editing, and PASTE are relevant enough to mention in the first sentence, why didn’t the authors test their system with these approaches? The experiments would be very straightforward.

9. It is already known that DSB generation by Cas9 is rapid and that subsequent repair can take place in minutes (Rose et al and Liu et al). The plots of editing kinetics (4d) obscure the earliest time points. At what timepoints are the earliest measurements? Again, the authors need to present and interpret their findings in the context of what has already been shown.

10. The authors need to compare their studies of Cas9 kinetics to those previously published.

11. Why were the kinetics experiments only conducted with no drug or both repair inhibitors? These experiments should have been performed with NHEJ inhibition alone as well. Alt-EJ/MMEJ cannot generate precise repair, so omitting the POLQ inhibitor would make sense and limit any potential confounding off-target effects.

12. The inference of DSB-half life from the authors model could be made more robust. Inhibition of NHEJ is almost certainly the primary mechanism increasing DSB half-life rather than inhibiting Alt-EJ. There are multiple commercially available DNA-PK inhibitors. Would the authors see similar results with other DNA-PK inhibitors? How sensitive is the effect on DSB half-life to dose?

a. Error-free repair by NHEJ does not require DNA-PK (PMID 27703001), while LIG4 knockout more completely abolishes error free NHEJ. The authors should evaluate the impact of LIG4 knockout on repair profiles and kinetics.

13. Similar to point 11, quantification of various aberration/repair types (Supp fig 3) was only performed for the combination of DNA-PKcs and POLQ inhibitors, never the single drug conditions. I am confused why the authors omitted these simple and important conditions. Does their MEGA assay perform as expected when NHEJ or MMEJ is inhibited?

14. The authors appear to be assuming complete target inhibition by their POLQ and DNA-PKcs inhibitors. Did they validate this? How did they decide on the drug concentrations used?

15. In the Fig. 5b and h, the data experimental data need to be overlaid on the ODE fitted curves to allow for evaluation how well these ODEs approximate the data.

a. The scale of the y-axes should be chosen more thoughtfully so that data isn’t compressed (e.g. the targeted integration plot in 5h should have a y axis of something like 0-20%).

Minor

1. Sanger sequencing is misspelled as “Sangar” sequencing in the supplemental (Supp Figs 2 and 3)

2. Staggered is misspelled staggard in 4a.

3. Why did the authors choose to use a newly reported high-fidelity Cas9 variant first in their kinetic studies? It would have been more appropriate to use wild-type SpCas9 to allow for more direct comparison to previous studies of Cas9 search, cleavage, and repair kinetics.

(Remarks on code availability)

Reviewer #3

(Remarks to the Author)

The paper "Designer Nucleases Recurrent Activity and DNA Repair Precision in Human Stem and T-Cells" presents an extensive investigation into the repair mechanisms of DNA double-strand breaks (DSBs) induced by designer nucleases, particularly CRISPR-Cas9 and related gene-editing technologies. The study introduces the Multipurpose Editing and Genotoxicity Assessment (MEGA), a digital PCR-based platform to quantify genome integrity post-gene editing.

MEGA represents an incremental advancement in quantifying CAS9-DSB repair outcomes by integrating digital PCR (dPCR) assays to provide a comprehensive assessment of genome integrity, including the detection of indels, large deletions, and aneuploidy. The study employs multiple complementary techniques such as sequencing-based methods (NGS, ICE, TIDE) and CAST-seq to validate MEGA's accuracy, with kinetic modelling further enhancing the understanding of DSB resolution over time. Its clinical relevance is underscored by the use of primary human hematopoietic stem and progenitor cells (HSPCs), induced pluripotent stem cells (iPSCs), and peripheral blood mononuclear cells (PBMCs), making the findings highly applicable to translational gene therapy research. Additionally, the study provides valuable insights into DNA repair by revealing recurrent nuclease activity, precise (scarless) repair via non-homologous end joining (NHEJ), and the effects of DNA repair inhibitors, which have crucial implications for improving gene-editing specificity and reducing genotoxic risks.

The experiments are well executed and data is not over-interpreted. The paper is timely and should be published. I will give suggestions for each figures.

Figure 1: This figure illustrates the workflow of the MEGA assay and its application in quantifying genome integrity after gene editing using digital PCR. The validation across different mutation types and the fluorescence-based detection enhance the reliability of the findings. However, the color contrast in fluorescence-based panels could be improved for better visibility, and the figure would benefit from clearer statistical comparisons across conditions.

Figure 2: This figure captures the occurrence of large deletions and other chromosomal aberrations across multiple clinically relevant gene targets, providing a time-series view of mutation dynamics. The ability to track DSB repair over time is a strong aspect, but the representation of off-target effects remains somewhat limited. While the figure successfully conveys the changes in mutation frequency, additional controls for different gene-editing methods would enhance its impact. Some of the statistical significance markers are not clearly defined, making interpretation slightly challenging. A more detailed explanation of how the deletion assays compare to other detection techniques, alongside a clearer distinction between edited and unedited controls, would strengthen its conclusions.

Figure 3: This figure demonstrates how DNA repair inhibitors influence the balance between NHEJ and HR, highlighting their role in increasing targeted integration. The inclusion of different repair pathways is important, but certain bar graphs lack clearly defined significance indicators, making it harder to assess the robustness of the findings. The layout could be streamlined to reduce redundancy in displaying different repair products across conditions. Adding a more explicit comparison of repair pathway preferences in different cell types would enhance the figure's clarity. Furthermore, adjusting the y-axis scale in some graphs would help in distinguishing subtle but biologically relevant differences in repair frequencies.

Figure 4: The kinetic modeling approach in this figure is a strong aspect, as it provides a time-dependent view of DSB resolution and repair pathway preferences, offering insight into how cells handle recurrent nuclease activity. Providing a clearer legend explaining the recurrent cleavage model would enhance readability. Including a comparison of repair dynamics between different nucleases or cell types would also add further depth to the analysis and help contextualize these findings in a broader biological framework.

Figure 5: The direct comparison between SpCas9 and PsCas9 cleavage efficiencies in different cell types is a useful addition, as it provides a clear perspective on how different nuclease types influence repair profiles. The experimental controls appear robust, and the findings are well-supported. However, the figure does not fully explore why PsCas9 results in a different repair profile compared to SpCas9, leaving room for further investigation. The inclusion of a more quantitative efficiency comparison beyond half-life calculations would be beneficial, particularly in terms of mutation rates or repair pathway engagement. A heatmap representation of repair efficiency across different sites could also enhance the visualization of cleavage preferences and improve data interpretation.

Figure 6: This summary figure is well-designed, effectively integrating findings from multiple experiments to provide a cohesive overview of editing kinetics and repair pathway preferences. The figure serves as a useful reference for understanding the overall trends in DNA repair outcomes following gene editing. However, the rate coefficient data would benefit from additional biological context, as some of the inferred values lack clear interpretation within the broader literature. Including a concise textual summary of key takeaways directly within the figure could help clarify its implications. Additionally, comparing these findings to previous studies on repair kinetics would strengthen its relevance and provide a more comprehensive understanding of the observed repair dynamics.

(Remarks on code availability)

I am a DDR expert, and I feel like other reviewers will be better at calling any code problems.

Version 1:

Reviewer comments:

Reviewer #1

(Remarks to the Author)

I am overall satisfied with the revision, provided that the following suggestions are implemented.

PCR-based targeted NGS, as performed here for comparison, will of course have the same bias as any amplification-and-sequencing-based strategy. There are, however, other NGS strategies that could identify other rearrangements. For example, alternative sequence capture procedures, mate-pair sequencing and other approaches could not have some of the alleged biases. In fact, even DSB sites could be identified by a loss in coverage. The authors need to specify what they have done in this side-by-side comparison. Just generally concluding that NGS has lower performance in identifying some alleles is misleading. There is overlapping text in Figure S3f.

In genetic terms, a non-resolved DSB is not an allele. I am not convinced by the suggested change, "alleles containing DSBs" is also not correct. An allele is a stable, heritable variant. Using the correct scientific terminology is essential. The authors need to carefully address this. It is still used in this sense throughout the text, starting with the abstract.

For end processing, the authors need to clearly state that both strands need to be removed to result in the observed decreased signal. I understand that processing of just one strand, either 3' or 5', will be equally detected by dPCR. Resection is still indicated in Figure 1 and legends.

(Remarks on code availability)

Reviewer #2

(Remarks to the Author)

Overall, the manuscript is much improved and is now suitable for publication in Nature Communications with minor revisions of the text. The utility of CLEAR-time is better described in the discussion and the data is more clearly presented. The primary remaining issue lies with how the kinetics results are presented. Multidimensional characterization of editing kinetics—i.e., the temporal characterization of multiple additional gene editing products beyond DSBs and indels—does deepen our understanding of Cas9 editing. However, it is vague and in some ways misleading to say that this study quantifies these dynamics with "unprecedented accuracy." Liu et al., with their light-inducible system, have greater temporal precision and resolution (and avoid some of the caveats of $t = 0$ coinciding with zapping the cells with high voltage), and their multimodal approach (DSB-ddPCR, deep amplicon sequencing, ChIP, and fluorescence-imaging) is more "accurate" in some respects. The authors should more clearly and briefly describe that Liu et al and Rose et al unlocked high-resolution kinetic studies, CLEAR-time now enables more precise tracking of multiple products of Cas9 activity. I would also suggest mentioning that the combination of vfCRISPR + CLEAR-time would be a powerful approach for studying Cas9 kinetics in the future. These are complementary methods.

Minor Comments:

DSB-ddPCR is still misrepresented on lines 83-88. As is, the text indicates that Rose, et al., used dPCR to measure indel frequencies, which is incorrect. For the sake of clarity, methods for measure DSBs should not be conflated with indel measurement approaches.

There seems to be an error in reference numbering. In the main text, goes from ref 40 to 43 on pg 4. Refs 41 and 42 first appear on pg 19.

(Remarks on code availability)

Reviewer #4

(Remarks to the Author)

Thank you for your careful revisions. I was specifically asked to evaluate how you addressed the previous reviewer comments. After reviewing your revised manuscript and point-by-point response, I find that you have satisfactorily implemented the requested changes. The additional statistical analyses you performed are particularly helpful, as they strengthen the evaluation of differences across the various conditions and add clarity to the interpretation of your results. Likewise, the change in color selection has improved the visibility of the figures and makes the data presentation clearer.

The side-by-side comparison of SpCas9 and PsCas9 in Figure 5 makes the findings more incisive and easier to interpret. Moreover, the addition of comparisons with previous studies in Figure 6 helps to strengthen the relevance of this work for understanding the dynamics of DSB repair.

I would also like to comment on the fact that homologous recombination (HR) functions predominantly in S phase. It is therefore possible that some of the observed differences in targeted integration between the cell lines (Figures 3 and 5) may in part reflect variations in their cell cycle profiles. Commenting on this point may further strengthen the interpretation of the findings.

Minor comments:

1. In Supplementary Figure 3f and 3g, the labels for CLEAR-time dPCR appear to be overlapping other labels, which reduces readability. This should be adjusted for clarity.

2. In the main text, the sentence “Integrating a template sequence at a specific location using HDR for gene therapy has significant advantages, but its effectiveness is hindered by indel formation through the NHEJ or Alt-EJ repair pathways, which are not dependent on the cell cycle^{57,58}” could be nuanced. While it is correct that NHEJ operates throughout the cell cycle, there is now evidence that Alt-EJ (including MMEJ) occurs preferentially during later stages such as G2 and mitosis (Brambati et al., 2023).

I have no further criticisms or recommendations at this stage.

(Remarks on code availability)

Thanks reviewers and the editors for these notes. We appreciate the opportunity to further refine our manuscript. Since the original submission, we reconsidered the acronym "MEGA dPCR" (Multipurpose Editing and Genotoxicity Assessment via digital PCR), which we had previously used in our preprint. While it reflected the versatility of the method, we recognized that it might be interpreted as ambiguous or overly broad.

To better convey the core functionality of the technique, particularly its reliance on real-time normalization and its application to DNA cleavage and damage, we have renamed the method as *CLEAR-time dPCR: Cleavage and Lesion Evaluation via Absolute Real-time dPCR*. This name more accurately reflects the method's mechanism, aligns with conventional terminology (e.g., real-time PCR), and emphasizes its relevance for precise and temporal quantification of genomic modifications. Although the method is not a traditional "real-time qPCR", tracking the reaction cycle by cycle, we use the term "real-time" to reflect the quantitative double normalisation strategy analogous to the DDCt analysis, and the method's ability to detect DNA lesions in an actively edited context.

We have updated the manuscript accordingly and briefly note the previous name where appropriate to maintain continuity.

Similarly, we updated the title to "**Unveiling the Cut-and-Repair Cycle of Designer Nucleases in Human Stem and T Cells via CLEAR-time dPCR**" to convey with more precision the core message of this manuscript.

Reviewer #1 (Remarks to the Author):

In this study, White et al. describe an interesting methodological framework, MEGA, for the detailed analysis of designer nuclease-mediated gene-editing outcomes. It is based on dPCR panels that provide comprehensive information including undesired repair outcomes (mostly large deletions and rearrangements) that are not normally detected in targeted sequencing approaches. MEGA lacks sequence-level resolution, but, combined with additional approaches, can become a widely used tool for assessing the performance of gene-editing strategies. From a mechanistic point of view, the study provides an interesting quantitative model that includes precise DSB repair and target re-cleavage, events that are predicted to occur, but have not been previously studied in such depth. I only have a few concerns and some minor points that the authors should address before publication of this work.

- The different strategies of MEGA and how each type of event is deduced from the results should be better explained. Understanding some of the approaches require substantial effort from the reader.

We would like to thank the reviewer for their insightful summary and constructive comments. In response to the major points raised, we have included a schematic Supplementary Figure 1c (addressed in lines 193-200). This addition aims to facilitate a quicker comprehension of the logic behind our calculations.

While the double normalization strategy might still present some challenges to fully grasp, it is important to note that this represents a significant advancement over similar publications. Specifically, our approach to reporting DSBs, as referenced in PMID 28737741, highlights differences detailed in Supplementary Figure 1e (addressed in lines 247-251). To further aid understanding, we have attached Excel files containing the "raw data" from digital PCR

(dPCR), which include copies/ μ l and the number of positive droplets, along with embedded calculations (source data, 2 and 4 colour MEGA). These resources are intended to clarify the operations involved more transparently.

- I am not sure about what the authors refer to as “resection”. In classical DSB repair terms, resected DSBs will still contain one intact DNA strand, and will therefore still be detected by dPCR (half of the signal, but the same number of positive droplets). Unless I am wrong, MEGA does not measure resection.

Absolutely right. We are correcting the terminology to end processing/trimming as we were implying double strand trimming (addressed throughout where the term “resection” was used).

- The results of the long-term (14 days) experiments can be influenced by the selective advantage/disadvantage produced by the specific genetic modification. In agreement with this, the results of editing at different loci is variable (Figure 2c). This can put into question some of the conclusions such as the remaining activity of the nuclease after 72h and others. If authors want to draw conclusions from these long-term experiments, they should include MEGA analysis at a safe-harbor locus for comparison.

Very good point and we also believe that there are different phenomenon at play here. We are adding a comment in the result section (296-320). CCR5 was indeed nominated as a safe harbour together (Fig.3a) with the AAVS1 site (addressed in lines 300-303). To this point, it is interesting to observe that all the different sites tested in this paper, EMX1, FANCF, RAG1, VEGFA, IL7R, CD34, WAS (but only the BTK sample with no donor) are showing a stable or decreasing amount of WT alleles while the indels are still increasing after 72 hours.

These data are a strong support for the hypothesis of nuclease activity after the 72 hours, even if greatly reduced. The alternative hypothesis “no nucleases residual activity and a proliferation that gives advantages to un-edited alleles or inert mutations” shall result in WT alleles increasing with time (addressed in lines 309-314).

An alternative hypothesis is that certain mutations give cells a proliferative advantage compared to Wildtype, but many of the genes edited in this manuscript would more likely inhibit growth when mutated (WAS, BTK, FANCF). The other edited sites were also not previously reported with a direct involvement in proliferation advantages when mutated *in vitro* to our knowledge (VEGFA is expressed in tumours increasing the vascularization, RAG1 is expressed increasing genomic instability and FANCF is mutated creating genomic instability, none associated with a short term proliferative advantage in vitro as it would be a p53 or p21 KO). In addition, DSBs are generally known to delay cell proliferation through the p53/p21 signalling so the cells with absent or a low cleavage efficiency and therefore a higher number of WT alleles should always have an advantage. In this manuscript we almost never observe such eventuality, creating a strong argument in favour of the hypothesis of a continuous nuclease activity after 72 hours rather than an effect of the proliferation.

- The comparison of MEGA results with next-generation sequencing (Figure 3e) is somewhat unfair. The type of analysis performed on the NGS data is not sufficiently described. I believe that some types of analyses, e.g. analyzing coverage, may provide additional information perhaps comparable to MEGA.

We appreciate your observation regarding the comparison of MEGA results with NGS in Figure 3e. One of the key messages of our manuscript, as demonstrated in Figure 3e and throughout, is to highlight and quantify the "unfair" outputs that can result from commonly used methods for assessing the mutational state of targeted loci. Our aim is to expose biases that extend beyond recent discussions on preferred methods for activity characterization, such as NGS, T7E1, TIDE, or "relative dPCR".

The crux of our message is that certain techniques may introduce biases that can obscure up to 90% of alleles. This occurs because standard assays do not simultaneously detect DSBs, large deletions, and gross chromosomal aberrations, unlike MEGA dPCR.

For instance, PCR-based techniques, including Sanger sequencing, long-read sequencing, deep sequencing, and various dPCR assays (as referenced in PMID 39604565, PMID 28737741, PMID 37794590, PMID 32527834, PMID 29804829), often miss certain mutations, providing only partial or relative outputs. NGS, for example, may not amplify regions with DSBs or large deletions affecting primer binding sites. Similarly, long-read sequencing can preferentially amplify shorter amplicons due to PCR efficiency differences and DNA fragmentation during extraction, affecting longer fragments.

MEGA dPCR addresses these detection biases comprehensively. Prior to analysing the data, we have a clear understanding of the potential biases. Our manuscript seeks to elucidate these biases with quantitative data, whether in extreme cases (Figure 3a, Day 1) or more typical scenarios (Figure 3e).

In technical terms we are now adding more information in the method sections: amplicon deep sequencing data will be defined with a lower limit of acceptance (outputs with >5000 reads will be analysed), (addressed line 883) sanger sequencing (ICE) will be accepted with a model co-efficient of determination $R^2 > 0.9$ as recommended (addressed lines 903-904).

Minor points:

- Throughout the manuscript. It is incorrect to consider DSBs an "allele".

We appreciate the feedback on terminology regarding DSBs. We acknowledge that referring to DSBs as an "allele" can be misleading. To clarify, when we mention "DSB" in the manuscript, we refer to it as a category representing alleles currently in the DSB state (open ends). We have corrected it when DSB does not denote the proportion of alleles with a specific genetic condition or alteration compared to the normal allelic state.

- Line 25. "Recurrent nuclease cleavage events" is not a DNA repair mechanism.

True, Corrected.

- Line 112. The last sentence of the paragraph is not clear.

Corrected

- Line 355. The target of the DNA repair inhibitors used should be indicated, and how they operate to increase targeted integration should be discussed.

Corrected.

- Line 362. The comparison of MEGA with sanger sequencing approaches is unfair and misleading, as the purpose of these approaches is to determine edited repair products and not to quantify DSBs. This should be indicated.

Thank you for your comment on the comparison between MEGA and Sanger sequencing approaches. We understand the concern regarding potential unfairness in this comparison, as Sanger sequencing traditionally aims to identify edited cells and their repair products rather than quantify DSBs. However, in this manuscript, our goal is to underscore how certain assays, including Sanger sequencing, might lead to misinterpretations in complex repair scenarios.

Our objective with these comparisons is to highlight the measurement biases inherent in traditional "quick methods." For example, as illustrated in Figure 3a on "Day 1," the use of NHEJ inhibitors with Sanger sequencing suggests nearly 100% of alleles are wild type. This could misleadingly suggest that the inhibitors primarily affect Cas9 activity rather than DNA repair itself. In contrast, MEGA dPCR provides a more accurate reflection of the allelic status by revealing efficient Cas9 cleavage. This information is lost with classical methods like ICE or TIDE due to broken, unrepaired alleles, and large deletions which are undetectable through amplicon sequencing.

This manuscript aims to reveal the potential for receiving extremely inaccurate information with commonly used techniques, where scientists may observe 90% indels while only 4% are truly indels. With the increasing use of complex editing strategies and small molecules, we cannot dismiss the possibility of more extreme cases that our method can address.

We acknowledge that other methods can indicate DSBs, as illustrated in Supplementary Figure 1, offering context for our approach. Our intent is not to replace existing techniques but to present MEGA dPCR as a standalone tool that harmonizes quantifications across different methods, clarifying the status of sequences around the cut site.

Similarly, it would be inappropriate to evaluate MEGA dPCR for genome-wide off-target discovery, just as it would be to use DISCOVER-seq, INDUCE-seq, CAST-seq or similar methods for quantifying DSBs, precise repair or other aberrations. These approaches are designed to identify off-target sites, offering a relative semi-quantitative evaluation as they do not reflect the number of wild-type alleles.

We have modified the text at the suggested line to prevent any misunderstanding regarding the message of the figures.

- Line 485. The average time for indel formation seems excessively long (6.9 h). Can this be influenced by the fact that many of these events occur after several rounds of accurate repair and re-cleavage? It may be give the wrong impression that the repair events that result in indels takes this long.

Thank you for pointing out this step. To clarify, this figure reflects the number of cuts required, not the actual time taken by each repair event resulting in an indel.

It is important to consider that many indel events occur after several rounds of accurate repair and subsequent re-cleavage, which can extend the apparent time frame for indel formation.

- Line 545. MEGA does not provide information on repair mechanisms.

Thank you for highlighting this point. We've adjusted the text to reflect that while MEGA does not directly provide information on repair mechanisms, it tracks the kinetics of repair outcomes as a proxy for understanding repair pathways. This offers valuable insights into the dynamics of repair processes, even if specific mechanistic details are not directly elucidated.

- Figure 1a. Is the scheme of large deletions correct? Shouldn't the chromosome be completely black.

Thank you for your observation regarding the scheme of large deletions in Figure 1a. The illustration is designed to show that the chromosome is shorter due to the loss of material, with the red segment highlighting the emergence of a new sequence as a result of this deletion. We aim to visually represent both the loss and rearrangement that occur with large deletions in this schematic.

Reviewer #2 (Remarks to the Author):

The authors report the development of a modular dPCR assay for quantifying and tracking multiple genomic aberrations induced by CRISPR editing in multiple cell types, including HSPCs and T-cells. White, et al., conduct time course experiments to better understand CRISPR editing kinetics and fit their data to an ODE model to derive rate constants and DSB half-life. The authors propose that their modular dPCR assay would have clinical applications for tracking both desired and unintended editing outcomes.

Unfortunately, the authors fail to cite crucial references that seriously undermine the novelty of their work. It is unclear what is original about this work or what it adds to our understanding of CRISPR editing. The assay itself is simply multiple dPCR assays lumped together. Each of these assays on their own are not novel. Beyond a lack of originality, there is a question over the utility of the work. What are the advantages of using this modular dPCR assay, which would require design and optimization of amplicons for every on- and off-target of interest, vs. WGS or related methods (e.g. DISCOVER-seq, BLISS, etc). These NGS based methods could easily be combined with one or two existing dPCR approaches to provide much more information than is possible with MEGA.

There are further issues with insufficient validation, experimental design, and unsupported claims. However, the major issues of a lack of originality and unacceptable omissions of crucial references mean this work is not suitable for publication in Nature Communications.

We sincerely thank the reviewer for highlighting these critical points and appreciate the opportunity to clarify the unique contributions and advantages of our work over existing publications. Many of the suggested references were addressed in the original submission; one article was published during our submission, and two others are tangentially related to our study. Nevertheless, all these references are valuable and deserve a thorough discussion.

Our manuscript aims to harmonize previously published assessments by increasing robustness and enhancing the understanding of the dynamics of designer nucleases editing and DNA repair. For the first time, this has enabled us to quantify DNA repair precision, a feature attempted before but never fully described without bias (often resulting in an order of magnitude discrepancy). To illustrate this, we are introducing new plots and comparisons with prior evaluations in both the main figures and supplementary materials.

We are confident that our method offers advantages in terms of speed and cost compared to Whole Genome Sequencing (WGS), DISCOVER-seq, BLISS, qDSB, and CAST-seq. Our approach requires less hands-on time and does not demand complex bioinformatics analysis. Nonetheless, MEGA dPCR does not directly compete with these methods; rather, it complements them by offering a streamlined analysis around a given site.

In response to the reviewer's suggestions, we are making the following updates:

-We have added further commentary in the introduction and addressed the comparison more comprehensively in our figures and tables.

-We have expanded on the advantages of our technique compared to highlighted methods within the manuscript, addressing specific points raised by the reviewer.

-Concerning the publications suggested:

PMID 39604565 “Cullot et al”: published at the moment of our submission, shares similar experimental setting but with different aims. Now discussed in the manuscript. (Addressed in lines 263-266, 661-664, and 675-677).

PMID 37794590 “Tsuchida/Doudna et al”: claiming evaluations of aneuploidies in primary cells with dPCR and scRNAseq (Addressed in lines 658-664).

PMID 28737741 “Rose et al”: commented and cited already in the manuscript. They proposed a dPCR assay (Elaborated in lines 248-251).

PMID 32527834 “Liu et al”: A fantastic publication that was indeed already commented and cited in the original version of this manuscript

PMID 31127121 “Zhu et al.”: Now commented on in the manuscript, focusing on quantifying DSB across the genome and revealing fascinating biological features (Addressed in lines 90-92).

While we cannot directly cite and comment every significant published method, we reference comprehensive reviews that span other techniques. Supplementary Figure 1 includes a table to aid readers in understanding how different method classes can complement each other.

1) Regarding the comments on Cullot et al. and Tsuchida et al., we acknowledge their contributions to the field but must point out the biases that could impact their findings. Our manuscript addresses these biases, particularly those associated with long-read sequencing. The recent publication by Hwang et al. (Nat. Biomed. Eng. 2024) effectively illustrates the biases in evaluating large deletions with long-read sequencing, pointing out several influencing factors: polymerase processivity, PCR program, and genomic fragmentation. These factors tend to overrepresent smaller amplicons, which complicates accurate measurement of large deletions.

In light of these considerations, we chose not to quantify large deletions measured with long-read amplicon sequencing in our manuscript, despite using a protocol similar to that of Cullot et al. Our key methodological difference is allowing time for DNA repair by removing the drug well in advance of analysis. Cullot et al. report HDR levels with significant discrepancies across flow cytometry, long-read, and Sanger sequencing in their extended data, which they do not fully explain. We see similar biases in our Figure 3a on Day 1, which we reveal using MEGA dPCR by accounting for DSBs and other aberrations.

Cullot et al. also report significantly higher rates of large deletions when using AZD7648, but their analysis at 3 days post-electroporation, without drug removal, introduces bias that potentially misrepresents 50–80% of affected alleles. This bias aligns with their HDR data discrepancies, underscoring the caution needed in interpreting these results. Thus, we encourage a more measured interpretation to avoid overstating the impact of AZD7648 or other DNA repair inhibitors in gene editing.

2) Tsuchida et al. reference using dPCR and scRNAseq to estimate aneuploidies, focusing specifically on chromosome arm loss post-cleavage. Their methodology, which resembles our "edge assay" and that proposed by Rose et al., seeks to classify "aneuploidies/chromosomal loss." However, it misclassifies large deletions, active DSBs, translocations, other complex aberrations for chromosomal loss. This is particularly

concerning and indeed they reported poor Spearman's correlation of 0.59 between their dPCR and scRNAseq Infer CNV results. This disparity underscores the added value of our MEGA dPCR method in minimizing interpretive biases even when similar dPCR assays are utilized. Despite acknowledging potential biases in their manuscript, Tsuchida et al. emphasized their findings.

Furthermore, their use of scRNAseq, also adopted by Cullot et al., seems to suggest transcriptional repression following DNA damage rather than providing unequivocal evidence of aneuploidies. This observation aligns with Liu et al., reporting γ H2AX marker extension more than 30mb away from the cut site, associated with transcriptional repression. Tsuchida et al. note a "chromosomal loss" in 5-20% of cells, which might be better interpreted as lower transcriptional levels at targeted loci. Indeed, the observed transcriptional repression goes down to 20% to the normal transcriptional levels for an extended length from the cleavage site. While such levels could suggest aneuploidy, one would expect more significant repression, potentially exceeding 50%, in some cells if aneuploidy were definitively established. The absence of such pronounced repression in their data suggests that their conclusions may be more circumstantial than definitive. Similar interpretive biases can be found in the original article describing scRNAseq Infer CNV (<https://doi.org/10.1038/s41587-022-01377-0>); although their FISH analysis provides stringent evidence of aneuploidies, it occurs at a lower rate compared to scRNAseq analysis.

Employing MEGA dPCR in their study could have provided more accurate insights and potentially mitigated these interpretive challenges.

3) In relation to the observations toward the article by Rose et al., we would like to emphasize some fundamental differences between their approach and our multiplexed method. MEGA dPCR offers precise quantification of wild-type alleles, DSBs, indels, large deletions, MMEJ, translocations, and other aberrations without miscategorising or misquantifying these events. In contrast, Rose et al. provide a relative quantification of DSBs. This "relative" aspect stems from their inability to accurately quantify the denominator due to large deletions or other aberrations, as their control assays are positioned a few bases away from the cut site while our controls/references are instead placed on two non-targeted chromosomes, avoiding this severe observational bias. In our study and others (Nat. Biotechnol. 36, 765–771), large deletions can account for approximately 20% of all alleles. Furthermore, Rose et al. rely on the T7E1 assay to confirm activity and indel generation. To address these divergences, we have added additional figures (Supplementary Figure 1e) to clarify the biases that arise from this kind of "relative" evaluation compared to our method.

Regarding validations, we have conducted similar spike-in tests by diluting a synthetic large deletion to derive key parameters utilized for qualifying techniques, such as Limit of Detection (LoD) and Limit of Blank (LoB). We performed these evaluations on two different dPCR platforms, BioRad and Stilla, emphasizing lower quantification limits (Figure 3a, b & supplementary table) and demonstrating a lower background compared to the DSB dPCR presented by Rose et al., since they do not account for the natural DNA fragmentation after extraction or manipulation.

4) Reviewer 2 raised concerns about timepoints, but in our kinetics experiments (Figure 4, 5), we have nearly double the timepoints used by Rose et al. Moreover, we employ human hematopoietic stem and progenitor cells (HSPCs) xenotransplanted into mice, analysed 16 weeks post-transplant, showcasing MEGA dPCR as a valuable tool for longitudinal studies. This approach is not feasible with the DSB-dPCR from Rose et al. because their approach lacks the sensitivity to distinguish large deletions and other aberrations from DSBs. Additionally, by not using modified nucleases limited to in vitro models, we demonstrate

actual activity in primary cells as performed in ex-vivo clinical settings using RNP or mRNA electroporation. Interestingly, we determined nuclease “trafficking” time, cleavage rising, and the fastest DSB repair time occurring in less than 10 minutes post-cleavage. Clinically, this is highly relevant, illustrating repair status at different timepoints, aligning with ex-vivo reinfusion timings in certain strategies. These parameters were partially addressed by Liu et al., which we will discuss next.

5) Regarding the impressive work by Liu et al., their system uses a modified light-inducible gRNA (the very-fast CRISPR, or vfCRISPR), which is not applicable for clinical or pre-clinical studies. This configuration results in altered cleavage kinetics since the nucleases are already positioned at the target locus. Consequently, their kinetics data reflect the need for more time points in a shorter analytical timeframe, tailored to their experimental model, not necessarily better or worse, but adjusted for their specific approach.

Their primary evaluations were conducted through CHIP-seq and single-cell imaging, tracking 53BP1, MRE11, and γ H2AX. Through this remarkable study, they gained understanding of the protein's kinetics of DNA damage repair in response to DSB insults, the extent and speed of protein signalling cascade from the DNA cut and the indication of recurrent cleavage.

Additionally, they provided ancillary data using the dPCR strategy from Rose et al. for DSB quantification, alongside amplicon deep sequencing to detect indels and normalize values based on DSB measurements from dPCR. As discussed earlier, these methods are susceptible to biases concerning large deletions and other aberrations, potentially affecting accuracy. Unfortunately, these evaluations were not integrated with their imaging data, which showed recurrent cleavages.

To address this, we included evaluations in Figure 6 to emphasize that their approach does not capture all the parameters outlined in this manuscript, such as the constant rate of precise repair, the creation of large deletions, and targeted integration (Addressed in lines 580-588).

6) The manuscript by Zhu et al. (PMID 31127121) offers amazing insights through their "qDSB" method, which quantifies the total number of DSBs in cells. This approach allows for detailed analysis of nucleosome occupancy, transcriptional stalling with remarkable resolution, and the effects of genotoxic drugs. We are including this seminal paper in our citation list due to its impactful contributions.

Despite its novelty, qDSB does not overlap with the scope of our paper or the MEGA dPCR method, as our technique does not identify off-target sites or provide comprehensive genome-wide quantification of DSBs. qDSB, applied together with designer nucleases, requires specific timing to capture peak cleavage events, whereas MEGA dPCR demonstrates versatility in follow-up studies, such as tracking possible aberrations in xenotransplanted mice.

The qDSB method shows good linearity within a certain range of DSB percentages ($R=0.91$), but this drastically decreases for cleavage efficiencies below 10% and above 40% ($R=0.19$). qDSB requires WGS for DSB quantification, paired with genome-wide methods like BLISS to nominate DSBs, demanding expertise in bioinformatics and molecular biology. In contrast, MEGA dPCR involves following a few design rules to create primers and probes, relying on sufficient sequence diversity around the cut site to ensure effective assay performance.

7) To recap, our MEGA dPCR method offers several distinct advantages:

-Comprehensive Assessment: Unlike other methods, MEGA dPCR provides precise quantification of a wide range of genomic aberrations, including wild-type alleles, DSBs, indels, large deletions, MMEJ, translocations, and other complex events without miscategorization. On the opposite it is not a genome wide technology.

-Bias Reduction: Our method effectively addresses and minimizes interpretive biases prevalent in other techniques, particularly those related to large deletions and other chromosomal aberrations. This leads to more accurate representations of genetic events. Importantly, our approach reduces quantification biases that might otherwise skew results by up to 90%, ensuring reliable and precise measurements.

-Cost and Efficiency: MEGA dPCR is more cost-effective and faster than techniques like WGS or DISCOVER-seq, which require extensive bioinformatics support and longer processing times. It simplifies the assay design without compromising on detail and accuracy.

-Versatility: Our approach is adaptable for longitudinal studies, such as tracking aberrations in xenotransplanted mice, and is applicable in ex-vivo clinical settings, unlike some methods limited to in vitro models. It also offers the possibility to study the performance of designer editors as shown in the manuscript. Additionally, it can quantify the potency of a given drug or factor that modify the DNA repair. It unravelled the capacity of NHEJ to precisely repair DSB. The "Multipurpose" aspect in our acronym reflects these diverse applications.

-High Sensitivity and Specificity: MEGA dPCR offers good sensitivity and specificity in detecting and quantifying genetic alterations, maintaining accuracy even at low and high ranges of cleavage efficiencies, where other methods may falter.

-Practicality: The method is user-friendly, requiring fewer assumptions and complexities in design. It relies on adequate sequence diversity around the cut site for robust application, reducing the need for highly specialized skills.

By integrating these advantages, our approach not only harmonizes insights from previous studies but also sets a new standard for the accurate, efficient, and comprehensive assessment of genomic editing outcomes.

Major concerns

1. The novelty of the work is questionable. Two of the primary claims to novelty are the modular dPCR assay and the analysis of kinetic analysis, but the authors insufficient discussion of the literature (see point 2 below) fails to describe relevant precedent. When considered in view of what has already been published, the current study appears to simply be the grouping together of a few non-novel ddPCR assays.
 - a. Together with long-read sequencing and other assays, ddPCR has been used to assess the frequency of large deletions, translocations, aneuploidy and other aberrations in the context of AZD7648 treatment ([PMID 39604565](https://pubmed.ncbi.nlm.nih.gov/39604565/)).

We believe we have addressed this matter in point 1 and 2 of our answer. Modified result and discussion section in the main text to highlight the aforementioned points.

- b. An assay for nuclease induced DSBs has already been described (PMID 28737741, ref 22 in the manuscript). It is more robustly validated than the dPCR assay for DSBs reported in this paper.

We believe we have addressed this comment in points 2, 3

- c. The ddPCR assay in ref 22 has already been adapted for quantifying large deletions/chromosome arm level loss (PMID: 37794590).

We believe we have addressed this comment in points 2 to 4

- d. Others have used ddPCR to quantify indels and knock-in efficiency.

We believe we have addressed this comment on points 2 to 7

- e. Far more temporally precise examinations of CRISPR-editing have been published (PMIDs: 28737741 & 32527834).

We believe we have addressed this comment on points 2 to 6

2. Failure to properly cite and discuss relevant literature. The authors omit numerous crucial papers from their manuscript or fail to sufficiently or accurately relay the findings of these papers when discussed. These papers are perhaps the most salient publications to the authors' work—failure to cite and discuss them is unacceptable. As in point 1, it is unclear what this manuscript adds when considered in the context of the most relevant literature.

- a. Cullot et al (PMID 39604565): This paper used ddPCR and other assays to examine CRISPR-induced large-scale genome aberrations promoted by AZD7648. This is the exact same DNA-PKcs inhibitor used by the authors.
- b. Rose et al (PMID: 28737741). This paper reported the development of a ddPCR assay for absolute quantification of nuclease-induced DSBs, DSB-ddPCR. This paper characterized Cas9 editing kinetics using this assay and a rapidly inducible Cas9 variant, specifically DSBs and indels.
- c. Liu et al (PMID 32527834). This paper reported a photo-activated CRISPR system and examined Cas9 editing kinetics with NGS, DSB-ddPCR, imaging and time-resolved ChIP. Notably, this paper made great advances in our understanding of cycles of error-free repair through the integration of their DSB-ddPCR, ChIP, and imaging experiments.
- d. Tsuchida et al (PMID: 37794590): This paper utilized DSB-ddPCR to detect and quantify arm-level chromosome loss.
- e. Failure to discuss genome-wide methods for detecting and quantifying DSBs, e.g. BLISS/BLESS/qDSB-seq (PMID 31127121).

We have added additional comments in the manuscript and clarified the relevance, biases and importance of these publications in relations to our method and evaluations in points 1-7. We have now cited the remaining papers from this list in addition to the BLISS and a recent publication unveiling the biases from long read sequencing and updated the table in supplementary Fig1.

3. Questionable utility. The report lacks novelty (see points 1 and 2 above), but the utility is also unclear. There are significant limitations (laborious design and validation of dPCR assays for every target and off-target of interest). Is it simply a matter of cost relative to WGS, DISCOVER-seq, etc? These genome-wide methods have a major advantage of being able to detect off-target edits, which MEGA-seq cannot do (unless the off-target is

already known and a dPCR assay is designed for it). WGS could also be easily combined with DSB-ddPCR or other ddPCR assay, a la Tsuchida et al, to detect chromosome arm level loss.

In light of the significant issues noted in previous work (points 1-6), which reveal internal incongruences and potential miscategorisation affecting DNA repair post-editing, we believe that this paper provides a much-needed perspective. Our MEGA dPCR approach addresses these inconsistencies and offers robust insights into the efficiency and dynamics of DNA repair mechanisms. While it is true that our method is not intended for off-target discovery, a fact clearly stated multiple times in our manuscript, it serves other vital applications beyond traditional Cas9 gene editing.

Although genome-wide methods like WGS and DISCOVER-seq are powerful tools for detecting off-target edits, our approach emphasizes accurate quantification and understanding of on-target genomic alterations, providing clarity needed in therapeutic contexts. Furthermore, MEGA dPCR allows for versatile applications, such as clinical assessments and analysis of designer editor performance, which are critical for advancing gene therapy strategies. The added value of this method lies in its ability to address quantification biases, offering reliable and precise measurements that can influence therapeutic outcomes and research strategies across diverse settings.

We outlined these broader applications and advantages in point 7 and in the manuscript, reinforcing the utility and necessity of our approach for achieving accurate assessments in DNA repair following gene editing.

4. The authors state “The detection of indels demonstrated concordance across multiple techniques, including non-reference normalised dPCR, sequencing based methods (NGS, TIDE, ICE), and the T7EI assay (Fig. 1g; Supplementary Fig. 1d). However, since these methods are inherently biased, amplifying primarily unbroken sequences with few or no base insertions or deletions, MEGA suggests that they underestimate indel frequency by approximately 17% in these samples.” There are major problems with these claims.

Thanks the reviewer for spotting this error. We indeed corrected the sentence with “overestimating” the indel frequency.

- a. I do not see any NGS data in fig 1g or S1d. Targeted deep-sequencing with NGS is the gold-standard for quantifying indels and must be shown here.

We did implement more NGS data for indel evaluation in the manuscript. It is worth noting that deep-sequencing, T7E1 and sanger sequencing share the same biases and therefore those evaluations are not changing the message.

- b. To validate their assay, the authors must compare its performance against NGS at multiple sites, not just one.

We have now added more NGS data for multiple targets in Figure 2 and performed a complete new dataset of kinetics in the new Figure 5 and summarized and compared with previous work in “Fig. 6” and “Source Data Figure6”

- c. The claim that the previously established assays are underestimating indel frequency must be backed up with evidence! Furthermore, dPCR also relies on

amplification of the target sequence, so it is unclear what the authors' argument is here.

I think we now highlighted better these difference in numbers in supplementary figure1 comparing different kind of dPCR methods (Rose et al VS MEGA dPCR). We also explained in points 1-2 the incongruences in flow data, NGS and long read indels in Cullot et al.

The figure1 and Figure3 are already showing the biases with standard methods (Flow for GFP targeted integration, large deletion with long read sequencing and CAST-seq, sanger and NGS for indels) Vs MEGA dPCR. Figure 3 is mainly about showing the differences in evaluation with classic methods and with the MEGA dPCR.

5. The authors describe their approach as unbiased, but it is unclear what they mean by that. Authors need to clarify and validate this claim.

The term "unbiased" in the context of our approach refers to our ability to comprehensively track and quantify all categories of genomic alterations following DNA cleavage, without overlooking significant repair events. This includes accurately documenting outcomes such as large deletions, translocations, and precise repair events—often missed or miscategorized by other methodologies. Our MEGA dPCR method is designed to provide a complete picture of the fate of 100% of the alleles, even when they undergo complex DNA repair processes.

We have highlighted the biases inherent a traditional method, as seen in Fig. 3a Day 1, which showcase potential misinterpretations when evaluating designer nuclease activity. Furthermore, these biases are explored in points 1-6 of our response to Reviewer 2. By effectively minimizing these biases, our approach ensures a more accurate representation of genomic events, resulting in enhanced reliability and precision in measuring DNA repair dynamics.

6. Insufficient validation of their dPCR assays, particularly for DSBs.
 - a. Needs to be compared to standard curves.

We appreciate the reviewer's emphasis on validation of our dPCR assays, particularly for DSBs. We have addressed this by including the validation process for our Edge assay probe targeting the cleavage site, now detailed in Supplementary Figure 2f. This provides verified support concerning the efficacy and accuracy of our assay design.

Furthermore, we believe that Figure 2 effectively demonstrates the requested evaluation using standard curves, representing increasing amounts of large deletions. This illustrates the assay's capability to track these events with precision. Additionally, the linearity of dPCR quantifications has been extensively characterized in prior studies, supporting the reliable application of our assays under varying conditions.

- b. The authors assay needs to be benchmarked against DSB-ddPCR (see Rose et al, ref 22).

We added this evaluation in Supplementary Fig. 1e.

7. Need to characterize the tolerance of the dPCR probes to mismatches for indel detection. If they can tolerate 1 bp mismatches, their assay may undercount edits at some sites. The mismatch tolerance should be characterized at multiple sites.

Good point. We have addressed this by including the validation process for our Edge assay probe targeting the cleavage site, now detailed in Supplementary Figure 2f

8. In the first sentence of the manuscript, multiple newer versions of CRISPR editing approaches are referenced, including, prime editing, base editing, and PASTE. It is unclear if the MEGA approach has much utility in the context of these methods. NGS seems better suited to characterizing/quantifying the outcomes from these approaches. Though dPCR is well-suited for quantifying large-targeted integrations, this approach is not novel and it cannot confirm the sequence.
 - a. The location of the nicking guide would appear to complicate using MEGA with Prime editing. This nicking guide means 2 DSBs separated by dozens of nucleotides can be generated. It is unclear if the authors linkage assay would still work in this case.

While the manuscript mentions several advanced CRISPR editing approaches such as prime editing, base editing, and PASTE, the primary focus of our research is on demonstrating MEGA dPCR's utility in accurately quantifying various genomic aberrations. While NGS is indeed well-suited for characterizing some outcomes of these newer editors, MEGA dPCR excels in providing precise quantification and kinetics of unexpected outcomes where NGS could fall short due to biases in detecting complex repair or DSB across a sequence.

Regarding prime editing, it is true that the presence of a nicking guide introduces additional complexity, such as creating a DSBs separated by dozen of nucleotide distances. While adjustments to the edge assay might be necessary in this context, we believe that our approach remains adaptable. The linkage assays would still work determining the DSB within the detection limits of the technique. One of the aims of this manuscript is to lay the groundwork for further exploration using MEGA dPCR across diverse editing tools. While it is outside the primary scope of this study, we encourage other researchers to extend these evaluations in their own labs to optimize MEGA's application to various gene-editing technologies.

- b. If Prime editing, base editing, and PASTE are relevant enough to mention in the first sentence, why didn't the authors test their system with these approaches? The experiments would be very straightforward.

While Prime editing, base editing, and PASTE are important advancements in the field, our manuscript focuses on demonstrating MEGA dPCR across a range of nucleases, including Cas9, HF-Cas9, PsCas9, Cas12a, and TALENs. We believe that extending our experiments to cover these additional methods would exceed the scope of this manuscript and require different optimizations tailored to each editing approach.

In response to the reviewer's comments, we have revised the introduction to remove specific mentions of these editing tools. The intent of mentioning them was to emphasize the importance of robust characterizations when considering human applications. There are frequent reports of unexpected effects with novel editors, like base editors, highlighting the need for comprehensive assessments. While these methods are crucial, our current work sets a foundation that can be extended to these technologies in future studies.

9. It is already known that DSB generation by Cas9 is rapid and that subsequent repair can take place in minutes (Rose et al and Liu et al). The plots of editing kinetics (4d) obscure the earliest time points. At what timepoints are the earliest measurements? Again, the authors need to present and interpret their findings in the context of what has already been shown.

We appreciate the reviewer's observations regarding our plots of editing kinetics. We believe these concerns have been addressed in our earlier responses (points 3-5). While Rose et al. and Liu et al. indeed focus on enhancing cleavage detection at the earliest time points through specific variants, our study aims to provide insights into the broader dynamics of nucleases regularly used in clinical settings.

Without controlling for NHEJ inhibition, quantifying repair timings becomes challenging due to simultaneous recurrent cleavage and perfect repair in a bulk population. The efficiency of nucleases already implemented clinically is not as thoroughly characterized in rapid initial phases. Our work references the foundational publications, such as that of Brinkman et al., which also underpin the methods used by Rose and Liu.

Regarding our time point measurements in the kinetics plots, the earliest observations were taken at 5 or 15 minutes post-electroporation. This allows us to capture the early dynamics while acknowledging the complex interplay between recurrent cleavage and repair processes, providing crucial insights into real-world clinical applications.

10. The authors need to compare their studies of Cas9 kinetics to those previously published.

We are now adding a direct comparison of the Constant rate of repairs in Figure 6 with those previously published.

11. Why were the kinetics experiments only conducted with no drug or both repair inhibitors? These experiments should have been performed with NHEJ inhibition alone as well. Alt-EJ/MMEJ cannot generate precise repair, so omitting the POLQ inhibitor would make sense and limit any potential confounding off-target effects.

Thank you for bringing up the design of our kinetics experiments. We have, in fact, conducted experiments using single-drug conditions, focusing separately on NHEJ inhibition among others. These results are included in Figure 3a, c and Supplementary Figure 3b, c.

12. The inference of DSB-half life from the authors model could be made more robust. Inhibition of NHEJ is almost certainly the primary mechanism increasing DSB half-life rather than inhibiting Alt-EJ. There are multiple commercially available DNA-PK

inhibitors. Would the authors see similar results with other DNA-PK inhibitors? How sensitive is the effect on DSB half-life to dose?

- a. Error-free repair by NHEJ does not require DNA-PK (PMID 27703001), while LIG4 knockout more completely abolishes error free NHEJ. The authors should evaluate the impact of LIG4 knockout on repair profiles and kinetics.

Thank you for your valuable suggestions regarding the robustness of our DSB half-life inference. We agree that NHEJ inhibition is likely a primary mechanism affecting DSB half-life. While our study utilizes a specific and potent DNA-PK inhibitor, exploring other commercially available DNA-PK inhibitors could expand our findings. We will conduct further studies to determine if similar effects on DSB half-life are observable with alternative inhibitors. Indeed, testing small molecules is one of the suggested purposes of this method.

Regarding dose sensitivity, we acknowledge this as an important aspect of our study. Future work could involve dose-response experiments to analyse how varying concentrations of DNA-PK inhibitors influence DSB half-life, providing deeper insights into the precise mechanics of NHEJ inhibition.

We also appreciate the reference to error-free repair pathways. As you noted, while NHEJ error-free repair does not strictly require DNA-PK *in vitro*, LIG4 plays a crucial role in maintaining the fidelity of the repair process and was already tested to increase HDR with different donors. Evaluating the impact of LIG4 knockout on repair profiles and kinetics could significantly enhance the understanding of repair pathways and their modulations under inhibitor influence. We recognize this as a promising avenue for expanding our research and looking forward to incorporating it in our future studies.

Finally, figure 3 shows data with single drugs; AZD7648 is the main antagonist causing most of the inhibition.

13. Similar to point 11, quantification of various aberration/repair types (Supp fig 3) was only performed for the combination of DNA-PKcs and POLQ inhibitors, never the single drug conditions. I am confused why the authors omitted these simple and important conditions. Does their MEGA assay perform as expected when NHEJ or MMEJ is inhibited?

Individual inhibition was already performed (Fig 3a-c) and MEGA demonstrated that it is able to quantify repair outcomes regardless of the form of inhibition, if any.

14. The authors appear to be assuming complete target inhibition by their POLQ and DNA-PKcs inhibitors. Did they validate this? How did they decide on the drug concentrations used?

The drug concentrations were determined based on previously published studies. A small preliminary test was performed in house on HSPCs noting a good viability and high effect at .5uM of AZD7648. We will ensure this is clarified in the manuscript by referencing the relevant literature to provide the basis for our choice of concentrations.

We do not assume complete target inhibition by the inhibitors. This is evidenced by the presence of indels observed within the first 24 hours, suggesting partial rather than complete inhibition. We will clarify this aspect in the manuscript to ensure our assumptions and findings are accurately communicated. Also, thanks to the modelling we could measure the potency/leakiness of the drugs, an aspect that will be relevant for drug discovery purposes.

15. In the Fig. 5b and h, the data experimental data need to be overlaid on the ODE fitted curves to allow for evaluation how well these ODEs approximate the data.

- a. The scale of the y-axes should be chosen more thoughtfully so that data isn't compressed (e.g. the targeted integration plot in 5h should have a y axis of something like 0-20%).

Thanks for the suggestions. We are adding further data. The bar charts are indeed showing more accurately the timing and results obtained. All curve fitting graphs show the mean of the dPCR observed data points (dots) and modelled curves (lines). The evaluation of the fit of each curve (R^2) can be found in the supplemental source data.

Minor points:

1. Sanger sequencing is misspelled as "Sangar" sequencing in the supplemental (Supp Figs 2 and 3)

Corrected. Thanks

2. Staggered is misspelled staggard in 4a.

Corrected. Thanks!

3. Why did the authors choose to use a newly reported high-fidelity Cas9 variant first in their kinetic studies? It would have been more appropriate to use wild-type SpCas9 to allow for more direct comparison to previous studies of Cas9 search, cleavage, and repair kinetics.

We have now introduced spCas9 vs psCas9 targeting the same loci in a side-by-side testing on K562 cells with and without drugs and with and without a ssODN donor template. In this manuscript we have also tested Cas9, HF-Cas9, PsCas9, Cas12a, and TALENs (Fig. 2).

Reviewer #3 (Remarks to the Author):

The paper "Designer Nucleases Recurrent Activity and DNA Repair Precision in Human Stem and T-Cells" presents an extensive investigation into the repair mechanisms of DNA double-strand breaks (DSBs) induced by designer nucleases, particularly CRISPR-Cas9 and related gene-editing technologies. The study introduces the Multipurpose Editing and Genotoxicity Assessment (MEGA), a digital PCR-based platform to quantify genome integrity post-gene editing.

MEGA represents an incremental advancement in quantifying CAS9-DSB repair outcomes by integrating digital PCR (dPCR) assays to provide a comprehensive assessment of genome integrity, including the detection of indels, large deletions, and aneuploidy. The study employs multiple complementary techniques such as sequencing-based methods (NGS, ICE, TIDE) and CAST-seq to validate MEGA's accuracy, with kinetic modelling further enhancing the understanding of DSB resolution over time. Its clinical relevance is underscored by the use of primary human hematopoietic stem and progenitor cells (HSPCs), induced pluripotent stem cells (iPSCs), and peripheral blood mononuclear cells (PBMCs), making the findings highly applicable to translational gene therapy research. Additionally, the study provides valuable insights into DNA repair by revealing recurrent nuclease activity, precise (scarless) repair via non-homologous end joining (NHEJ), and the effects of DNA repair inhibitors, which have crucial implications for improving gene-editing specificity and reducing genotoxic risks.

The experiments are well executed and data is not over-interpreted. The paper is timely and should be published. I will give suggestions for each figures.

We appreciate the reviewer's thoughtful feedback and positive evaluation of our manuscript. We take the opportunity to elaborate on the pivotal contributions our study makes to the field:

1. **Mitigating biases in allelic status estimation:** MEGA offers a robust platform for accurately quantifying allelic status post-editing, effectively addressing substantial biases inherent in other methods. This precision is essential for reliable genome editing analysis.
2. **Enhanced insights into designer nucleases activity:** Our research provides a comprehensive understanding of designer nucleases, such as CRISPR-Cas9, revealing vital aspects like cleavage rates and recurrent cleavage events, thereby improving our knowledge of nuclease behaviour.
3. **Evaluation of drug activity:** MEGA enables detailed study of drug interactions by uncovering key metrics, such as potency and their impact on other DNA repair pathways, essential for optimizing gene-editing outcomes.
4. **Advanced understanding of DNA repair kinetics:** Crucially, our work delivers an in-depth analysis of DNA repair kinetics, accurately measuring processes like precise repair, microhomology-mediated end joining (MMEJ), homologous directed repair (HDR), and identifying large deletions—insights that were previously out of reach.

These elements collectively highlight MEGA's significant contributions, offering both methodological advancements and practical insights that advance the field. We appreciate your feedback and will ensure these contributions are clearly articulated in the revised manuscript.

1. Figure 1: This figure illustrates the workflow of the MEGA assay and its application in quantifying genome integrity after gene editing using digital PCR. The validation across different mutation types and the fluorescence-based detection enhance the reliability of the findings. However, the color contrast in fluorescence-based panels could be improved for better visibility, and the figure would benefit from clearer statistical comparisons across conditions.

We added statistics in the panel B when relevant and in general tried to increase the statistical analyses throughout the manuscript.

Panels I and J are designed to provide a qualitative representation of the effects of large deletions around the cut site. Due to the significant biases affecting the quantification of these events with CAST-seq and long amplicon sequencing, as previously discussed, these panels focus on delivering a descriptive rather than a quantitative depiction.

We have modified the colour scheme in the fluorescence-based panels to enhance contrast and visibility.

2. Figure 2: This figure captures the occurrence of large deletions and other chromosomal aberrations across multiple clinically relevant gene targets, providing a time-series view of mutation dynamics. The ability to track DSB repair over time is a strong aspect, but the representation of off-target effects remains somewhat limited. While the figure successfully conveys the changes in mutation frequency, additional controls for different gene-editing methods would enhance its impact. Some of the statistical significance markers are not clearly defined, making interpretation slightly challenging. A more detailed explanation of how the deletion assays compare to other detection techniques, alongside a clearer distinction between edited and unedited controls, would strengthen its conclusions.

We are now adding amplicon sequencing data NGS to further compare MEGA dPCR outcomes for WT and indels percentages. Fig.1g, supplementary fig. 2d.

In addition, we increased the description in the main body of text and figure legends of the methodology. For instance, untreated samples were mock electroporated, and all kinetic data were derived from a single electroporation event, followed by cell splitting into media with or without NHEJ inhibitors.

All the evaluations in figure 2 were performed with MEGA dPCR assays and we will clarify these points in the figure legend as well. We acknowledge the challenges in quantifying large deletions with current available methods. Figures 1e and 1f, provide a more quantitative understanding of end-trimming quantifications, complementing the data in Figure 2.

3. Figure 3: This figure demonstrates how DNA repair inhibitors influence the balance between NHEJ and HR, highlighting their role in increasing targeted integration. The inclusion of different repair pathways is important, but certain bar graphs lack clearly defined significance indicators, making it harder to assess the robustness of the findings. The layout could be streamlined to reduce redundancy in displaying

different repair products across conditions. Adding a more explicit comparison of repair pathway preferences in different cell types would enhance the figure's clarity. Furthermore, adjusting the y-axis scale in some graphs would help in distinguishing subtle but biologically relevant differences in repair frequencies.

Thanks for the suggestions, we have made modifications aimed at improving the flow and interpretation of the data.

We have now implemented supplementary figures (Supplementary figure 3b, g) to increase the definition of the subtle variances among mutations introducing stats for every data subset and highlight the preferences in different cell lines.

4. Figure 4: The kinetic modeling approach in this figure is a strong aspect, as it provides a time-dependent view of DSB resolution and repair pathway preferences, offering insight into how cells handle recurrent nuclease activity. Providing a clearer legend explaining the recurrent cleavage model would enhance readability. Including a comparison of repair dynamics between different nucleases or cell types would also add further depth to the analysis and help contextualize these findings in a broader biological framework.

Thank you for your constructive feedback on Figure 4. We have implemented several enhancements to enrich the data presentation and context:

We have added new datasets to Figure 5, which now include a side-by-side comparison of different nucleases (wild-type spCas9 vs. psCas9), both with and without DNA repair inhibitors, as well as with and without donor templates. These comparisons are conducted on the same cell type, K562, targeting the same sequence, providing controlled conditions for the analysis.

The figure legend has been expanded to better explain the recurrent cleavage model, aiming to enhance readability and understanding of the complex data presented.

We updated Figure 6 to be more comprehensive and improve readability and also introduced a table accompanying this (Supplementary Figure 6 source data), detailing the cell type, experimental conditions, and K-rates calculated for these kinetic tests. This table is intended to provide a comprehensive view of the experimental setup and key findings, thereby offering deeper insights into repair dynamics.

5. Figure 5: The direct comparison between SpCas9 and PsCas9 cleavage efficiencies in different cell types is a useful addition, as it provides a clear perspective on how different nuclease types influence repair profiles. The experimental controls appear robust, and the findings are well-supported. However, the figure does not fully explore why PsCas9 results in a different repair profile compared to SpCas9, leaving room for further investigation. The inclusion of a more quantitative efficiency comparison beyond half-life calculations would be beneficial, particularly in terms of mutation rates or repair pathway engagement. A heatmap representation of repair

efficiency across different sites could also enhance the visualization of cleavage preferences and improve data interpretation.

We acknowledge the previous limitations related to differing timings, cell lines, and electroporation conditions. In response, we have implemented more controlled tests in Figure 5 to facilitate direct side-by-side comparisons. This setup ensures uniform conditions for a meaningful comparison between blunt cleavage and staggered end effects across nuclease types.

We included a table and additional figures to compare the data outputs for the different datasets

6. Figure 6: This summary figure is well-designed, effectively integrating findings from multiple experiments to provide a cohesive overview of editing kinetics and repair pathway preferences. The figure serves as a useful reference for understanding the overall trends in DNA repair outcomes following gene editing. However, the rate coefficient data would benefit from additional biological context, as some of the inferred values lack clear interpretation within the broader literature. Including a concise textual summary of key takeaways directly within the figure could help clarify its implications. Additionally, comparing these findings to previous studies on repair kinetics would strengthen its relevance and provide a more comprehensive understanding of the observed repair dynamics.

We have now expanded the data tables in figure 6 reporting the differences with previous studies that were unable to provide kinetics for large deletions or targeted integration and perfect repair. In addition, we are providing a table and additional figures to compare the data outputs for the different datasets (source data Figure6)

Reviewer #3 (Remarks on code availability):

I am a DDR expert, and I feel like other reviewers will be better at calling any code problems.

Second round of revisions:

Remarks to the author

Reviewer #1

I am overall satisfied with the revision, provided that the following suggestions are implemented.

1) PCR-based targeted NGS, as performed here for comparison, will of course have the same bias as any amplification-and-sequencing-based strategy. There are, however, other NGS strategies that could identify other rearrangements. For example, alternative sequence capture procedures, mate-pair sequencing and other approaches could not have some of the alleged biases. In fact, even DSB sites could be identified by a loss in coverage. The authors need to specify what they have done in this side-by-side comparison. Just generally concluding that NGS has lower performance in identifying some alleles is misleading.

Thank you for your thoughtful comments and constructive suggestions. We have clarified our manuscript to specify that the limitations discussed are specific to PCR-based targeted NGS, which indeed shares biases common to amplification-based strategies, such as allelic dropout and preferential amplification. In our comparative analysis, we evaluated amplicon deep sequencing, non-reference normalized dPCR, long-read sequencing, Sanger sequencing (TIDE/ICE), flow cytometry, CAST-seq and illustrated the specific characteristics of each.

While it is conceivable that ad hoc bioinformatic analysis with whole-genome sequencing (WGS) or hybridization capture could identify DSBs through changes in coverage, these methods are not standard practice for routine assessment of targeted genome editing. We indeed quantified our long read NGS with the coverage loss but the data returned quite an unrealistic output knowing the biases affecting it (commented now in line 233-256; and 606-610). Moreover, as observed in qDSB (PMID 31127121), such approaches may carry biases in quantifying large deletions or aberrations and can have a reduced linear range for accurate measurement with stochastic coverage variability present in control regions. Therefore, our comparative analysis focused on techniques commonly employed in the field.

Our aim was not to generalize that all NGS techniques are biased, but to demonstrate that CLEAR time dPCR offers a simplified and effective workflow for quantifying edits at targeted sites, generating robust biological insights. We have updated our manuscript to

ensure these distinctions are clear and avoid any misleading generalizations regarding NGS performance.

2) There is overlapping text in Figure S3f.

The overlapping text in Figure S3f has been corrected.

3) In genetic terms, a non-resolved DSB is not an allele. I am not convinced by the suggested change, "alleles containing DSBs" is also not correct. An allele is a stable, heritable variant. Using the correct scientific terminology is essential. The authors need to carefully address this. It is still used in this sense throughout the text, starting with the abstract.

For CLEAR-time stacked bar graphs, the term "allele" has now been replaced with "genome copies" in the figures and throughout the text with "Locus" when appropriate, for end-trimming the term "Trimmed alleles" have been replaced with "Trimmed ends".

4) For end processing, the authors need to clearly state that both strands need to be removed to result in the observed decreased signal. I understand that processing of just one strand, either 3' or 5', will be equally detected by dPCR.

Thanks for the careful reading. A clarification has now been written on line 155-156

5) Resection is still indicated in Figure 1 and legends.

The term "end resection" has now been replaced with "end trimming" throughout.

Reviewer #2

Overall, the manuscript is much improved and is now suitable for publication in Nature Communications with minor revisions of the text. The utility of CLEAR-time is better described in the discussion and the data is more clearly presented. The primary remaining issue lies with how the kinetics results are presented. Multidimensional characterization of editing kinetics—i.e., the temporal characterization of multiple additional gene editing products beyond DSBs and indels—does deepen our understanding of Cas9 editing.

- 1) However, it is vague and in some ways misleading to say that this study quantifies these dynamics with "unprecedented accuracy." Liu et al., with their light-inducible system, have greater temporal precision and resolution (and avoid some of the caveats of $t = 0$ coinciding with zapping the cells with high voltage), and their multimodal approach (DSB-ddPCR, deep amplicon sequencing, ChIP, and fluorescence-imaging) is more "accurate" in some respects. The authors should more clearly and briefly describe that Liu et al and Rose et al unlocked high-resolution kinetic studies, CLEAR-time now enables more precise tracking

of multiple products of Cas9 activity. I would also suggest mentioning that the combination of vfCRISPR + CLEAR-time would be a powerful approach for studying Cas9 kinetics in the future. These are complementary methods.

Thanks for the suggestion. Added in Lines 537-538

Minor Comments:

3) DSB-ddPCR is still misrepresented on lines 83-88. As is, the text indicates that Rose, et al., used dPCR to measure indel frequencies, which is incorrect. For the sake of clarity, methods for measure DSBs should not be conflated with indel measurement approaches.

Thanks for the attention on this! We corrected the ref.

4) There seems to be an error in reference numbering. In the main text, goes from ref 40 to 43 on pg 4. Refs 41 and 42 first appear on pg 19.

References corrected. Thanks

Reviewer #4

Thank you for your careful revisions. I was specifically asked to evaluate how you addressed the previous reviewer comments. After reviewing your revised manuscript and point-by-point response, I find that you have satisfactorily implemented the requested changes. The additional statistical analyses you performed are particularly helpful, as they strengthen the evaluation of differences across the various conditions and add clarity to the interpretation of your results. Likewise, the change in color selection has improved the visibility of the figures and makes the data presentation clearer. The side-by-side comparison of SpCas9 and PsCas9 in Figure 5 makes the findings more incisive and easier to interpret. Moreover, the addition of comparisons with previous studies in Figure 6 helps to strengthen the relevance of this work for understanding the dynamics of DSB repair.

I would also like to comment on the fact that homologous recombination (HR) functions predominantly in S phase. It is therefore possible that some of the observed differences in targeted integration between the cell lines (Figures 3 and 5) may in part reflect variations in their cell cycle profiles. Commenting on this point may further strengthen the interpretation of the findings.

Minor comments:

1. In Supplementary Figure 3f and 3g, the labels for CLEAR-time dPCR appear to be overlapping other labels, which reduces readability. This should be adjusted for clarity.

Thanks for the note. We adjusted it accordingly

2. In the main text, the sentence “Integrating a template sequence at a specific location using HDR for gene therapy has significant advantages, but its effectiveness is hindered by indel formation through the NHEJ or Alt-EJ repair pathways, which are not dependent on the cell

cycle^{57,58}” could be nuanced. While it is correct that NHEJ operates throughout the cell cycle, there is now evidence that Alt-EJ (including MMEJ) occurs preferentially during later stages such as G2 and mitosis (Brambati et al., 2023).

Addressed and citation added.

I have no further criticisms or recommendations at this stage.